# Modeling and Verification of Asynchronous Systems Using Timed Integrated Model of Distributed Systems

**DOI:** 10.3390/s22031157

**Published:** 2022-02-03

**Authors:** Wiktor B. Daszczuk

**Affiliations:** Institute of Computer Science, Warsaw University of Technology, Nowowiejska str. 15/19, 00-665 Warsaw, Poland; wbd@ii.pw.edu.pl; Tel.: +48-22-234-78-12

**Keywords:** timed distributed systems, distributed system timed specification, deadlock detection, distributed termination, model checking, timed automata

## Abstract

In modern computer systems, distributed systems play an increasingly important role, and modeling and verification are crucial in their development. The specificity of many systems requires taking this into account in real time, as time dependencies significantly affect the system’s behavior, when achieving the goals of its processes or with adverse phenomena such as deadlocks. The natural features of distributed systems include the asynchrony of actions and communication, the autonomy of nodes, and the locality of behavior, i.e., independence from any global or non-local features. Most modeling formalisms are derived from parallel centralized systems, in which the behavior of components depends on the global state or the simultaneous achievement of certain states by components. This approach is unrealistic for distributed systems. This article presents the formalism of a timed integrated model of distributed systems that supports all of the mentioned features. The formalism is based on the relation between the states of the distributed nodes and the messages of distributed computations, called agents. This relation creates system actions. A specification in this formalism can be translated into timed automata, the most popular formalism for specifying and verifying timed parallel systems. The translation rules ensure that the semantics of T-IMDS and timed automata are consistent, allowing use of the Uppaal validator for system verification. The development of general formulas for checking the deadlock freedom and termination efficiency allows for automated verification, without learning temporal logics and time-dependent formulas. An important and rare feature is the finding of partial deadlocks, because in a distributed system a common situation occurs in which some nodes/processes are deadlocked, while others work. Examples of checking timed distributed systems are included.

## 1. Introduction

The formalism of the distributed systems specification was developed at the Institute of Computer Science, Warsaw University of Technology. The formalism, called timed integrated model of distributed system (T-IMDS, based on the earlier timeless IMDS [1,2]) has a set of general features that distinguish it from other formalisms:*timed specification and verification*: distributed systems are asynchronous in nature, and time dependencies may substantially change their observed features; as in the Russian fairy tale: the crane offered the heron a marriage; the heron initially refused, but later changed her mind and offered to charm the crane when he took offense …; as a result, they see each other often, but each of them is proud and refuses when the latter is ready to propose; the tale has no end, because its characters act exactly in counter-phase, they do not synchronize in any way; in particular, some deadlocks can disappear due to time dependencies, while others can arise.*communication duality*: expressing the distributed system in terms of nodes with their states and agents with their messages, emphasizing communication duality in distributed systems: message passing versus resource sharing; the system specification can be switched between two system views: node view and agent view, yet with both in a uniform structure;*locality*: the node’s action is executed based on the current situation of the node (without any association with other nodes); no event outside the node (except for messages sent to it) can affect the behavior of the node;*autonomy*: each decision regarding the execution of the actions of the node (including the choice between many possible actions) is taken autonomously; the only way to influence the behavior of a node is to send a message to it (the message may enable an action that was previously disabled); no order is established in the set of pending messages, such as a stack or queue, the node autonomously decides which message will initiate the next action;*asynchrony*: the synchronization of nodes with agents is hidden inside processes (in actions belonging to processes); therefore, processes are perceived as asynchronous from the outside; also the communication channels are asynchronous: the message is sent irrespective of the local situation of the target node, in particular, whether it is waiting for this message or performing completely different operations;*automated verification*: expressing essential features of the distributed system’s behavior using general temporal formulas [2], not related to the internal structure of the system being verified.

A large selection of timed specification and verification techniques are offered; some are mentioned in Section 2. This is why we extended our IMDS formalism to cover real-time dependencies.

Locality and autonomy are obvious features of distributed systems. Asynchrony is very much needed between cooperating distributed nodes, because synchronous behavior requires some knowledge of the global state, which is difficult to obtain in a distributed system (or which cannot be obtained at all). Imagine a distributed component, for example a Büchi automaton [3] or Zielonka’s automaton [4]. If such an automaton is to communicate with another, both of them must follow a common transition. Obviously, such behavior is impossible: how would the automaton learn that they have both reached matching states or enabled matching transition? This principle breaks asynchrony (two automata cannot synchronize on states or transitions), locality (non-local/global information of the state of the other component is needed), and autonomy (the decision about the component behavior must depend only on its history and its preferences, rather than on the state of other components). Therefore, synchronous actions or other forms of direct synchronization are unrealistic in the description of distributed system behavior. Please note that such models are wrongly called distributed in numerous articles (e.g., [4]).

Communication duality is often emphasized as a theoretical feature of distributed systems, for example, in comparing client-server and remote procedure call (RPC) models, but rarely do both aspects occur in a uniform environment. In [5], they are both described, but as separate and opposite models. We argue that those models can simply be different points of view of a single, uniform system. The image of the modeled system depends on the manner of action grouping, rather than being a substantial property of the system itself.

Finally, identifying partial (sometimes called local) deadlocks and checking the distributed termination with model checking techniques [6] requires the designer to have some knowledge of temporal logic, since features should be expressed in terms of the elements of the particular system under verification [7,8]. The reason for this is that the deadlock is often identified as a global state, with no outgoing transitions [9]. Alternatively, a temporal formula that guarantees that the system is stuck in a deadlocked state is applied [10]. In both solutions, only a total deadlock can be identified. Other techniques are used for partial deadlock identification, but typically in systems having a required structure [11,12] (for example, looping systems). However, a situation in which some processes are deadlocked while other processes work is usual in distributed systems and should be found in systems of arbitrary shape.

The contribution of this article is the introduction of the timed integrated model of distributed systems (T-IMDS), which overcomes all the above-mentioned limitations in the specification and verification of distributed systems.

The specification is real-time aware, which is not novel between formalisms, but is an important step in IMDS evolution.It is based on a relation between two pairs of a distributed node state and the distributed computation (called an agent) message. The input pair only triggers a system action and produces a new pair that simply waits for its occasion (the consecutive actions of the node and the agent need not even be enabled at this moment). This counters the requirements for synchrony and non-locality of actions. The description models asynchronous actions, in which nothing depends on the synchronous delivery of any items needed to fire the actions. Distributed components make their autonomous moves based only on their local elements, and they do not depend on any global or non-local features of the distributed system.Communication duality (client–server versus RPC) is incorporated in the model; the views are only the decompositions or cuts of the system, the given view is achieved by a specific grouping of the actions; however, the set of actions remains the same.Identification of partial deadlocks and the partial distributed termination, as well as total–general temporal formulas are defined for this purpose, related to the features of the formalism, rather than to the features of the given verified system. The developed verification algorithms (which are beyond the scope of this article) allow both timeless and timed verification. The compatibility with Uppaal timed automata allows for verification under the external verifier Uppaal [13,14].

The Dedan verification environment (deadlock analyzer) was developed based on IMDS. It offers:interface to the specifications of the distributed system, text or graphic,a simple built-in temporal verifier (based on the TempoRG verifier [15]) for checking models of small and medium systems,simulator,export to external verifiers: Spin [16] for LTL verification, NuSMV [17] for LTL/CTL verification, Uppaal for CTL/TCTL verification.

The behavior of a distributed system may be time-dependent, as system changes may take some time. As a result, the dependence between periods associated with individual system changes can influence deadlock situations. Timed Petri nets and timed automata (TA) are two of the most popular formalisms for expressing the behavior of time-limited systems (TA) [18,19,20]. While automata-based models, as well as IMDS, can be transformed to Petri nets [21], they are not so attractive in our opinion, because it is not easy to extract the processes from the specification. We use Petri nets for the structural analysis of a verified system and some kinds of model checking [22]. The temporal logic timed CTL (TCTL) is associated with TA [18,23]. Two types of transition: progress transitions and time transitions are introduced to describe system changes and the flow of time:*Progress transition* (or: transition): If the Boolean expression on the automaton’s transition is true, the progress transition (or transition) is executed. This expression is composed of integers and clock values. TA actions are symbols associated with transitions (not to be confused with IMDS actions, we will use the term *symbol* for TA actions). When two or more automata have the same action symbol, the transitions are synchronous. The transitions are otherwise executed in an interleaving way [24].*Timed transition* can be executed if all of the automata in the set have time invariants (relationships between clock values and integers) in their current locations. All the clocks are simultaneously shifted by the same value (not exceeding any invariant).

This article is organized as follows: Section 2 covers related work on timed varication formalisms, especially timed automata. Section 3 deals the basic timeless formalism: the integrated model of distributed systems, and verification rules in IMDS, which are the basis of extension to timed systems. Section 4 covers the Uppaal timed automata model, the target formalism to which timed IMDS is converted. The timed version of IMDS is described in Section 5. The formal translation of T-IMDS to UTA is presented in Section 6. Section 7 gives two examples of distributed timed systems and their verification in the Dedan/Uppaal environment. The work is concluded in Section 8. The Appendix B contains the proof of equivalence between the semantics of T-IMDS and its implementation in UTA.

## 2. Related Work: Timed Automata and Other Timed Formalisms

Since system changes may take time, the dependencies between durations associated with specific changes may play a role in deadlock situations. Timed Petri nets and timed automata [19,20,25,26] are two of the most prominent formalisms for expressing the behavior of time-limited systems. Other real-time modeling techniques include hybrid automata [27] and timed CSP [28]. For discrete time modeling, among others, DTG (durational transition graphs [29]) and EMLAN [30] are used.

Real-time CTL (RTCTL, having time constraints attributed to temporal operators [31,32]), quantitative CTL (QCTL, with unit-delay transitions [33]), timed CTL (TCTL, connected with timed automata [18]) and bounded real-time model checking [34] were also developed. Discrete-time model checking example formalisms include clocked CTL (CCTL, based on time intervals in discrete-time systems [35]) and discrete-time CTL (DTCTL for embedded systems [30]).

Timed automata (TA [18]) are related to Büchi automata [3], but with the addition of time constraints. Automata execute their transitions separately (in an interleaving manner [36]), except for transitions on shared symbols, which synchronize the automata. A set of real-time clocks is used to achieve time limitations. The clocks are variables, with values ranging in ℝ_≥0_, multiples of a *basic unit of time* (a *unit* in short). A clock value is a real number, yet we can watch it while its values are integers or between integers. For example, with two clocks, *x* and *y*, we can examine the system in a circumstance where *x* is 2 and *y* is between 0 and 1. Similarly visible is the relationship between clocks (as *x* < *y* or *x* < *y* + 1). Time limitations, which determine the values of clocks at which specific transitions may be fired (for example, *x* ≥ 1), are attributed to the transitions of timed automata. Time invariants can also be enforced on automata states (called locations in TA). The time invariant expresses, in terms of clock values, how long an automaton may stay in a given location, such as *y* < 3. On transition, the clocks may be reset.

The TA transitions are instantaneous; the clocks advance while the automata remain in their locations. As a result, there are two sorts of advancement in TA: executing *progress transitions* and passing of time, referred to as *timed transitions*. The articles [18,23] give a complete overview of timed automata and their semantics.

*Compound progress transitions* and *compound timed transitions* are used to define the semantics of a set of timed automata:If a Boolean expression on a progress transition (or simply: *transition*, also termed *action transition* [37]) outgoing from a *current* location of a timed automaton TA is fulfilled, the transition can be executed. This expression is made up of integer constants and clocks. A difference is only allowed when two clocks are utilized in the expression. If more than one transition (in the same automaton or different automata) can be executed in a set of TA, the choice is nondeterministic. Transitions are denoted by symbols known as *actions* (do not confuse with IMDS actions, we use the term *symbol* for TA action). When two or more automata have the same action symbol, the transitions are executed synchronously. Otherwise, the transitions interleave.The timed transition can be executed if all of the automata in the set have their time invariants fulfilled in their current locations. The execution is based on synchronously advancing all clocks (by the same, real value > 0). The smallest difference between the maximum value of a clock used in an invariant and the present value used in this invariant is the maximum value to which the clocks can be advanced. If the invariants *x* < 2 and *y* ≤ 3 are present, and the current clock values are *x* = 1.5 and *y* = 2.6, the highest value is 0.4, which advances *x* to 1.9 and *y* to 3.

The option is nondeterministic if both a progress and a timed transition are enabled. It is important to note that there may be an endless number of timed transitions if a timed transition is possible. If a timed transition of 1 unit is achievable, perhaps a transition of 0.9 units makes no difference. As a result, timed regions (equivalence classes) are introduced to the formalism, with integer limits. The greatest integer relative to the clock is the maximum value used. Given integer parts of all clocks and an equal sign of the fractional part of clock differences (0 is treated as a distinct, third sign), a region, such as ⎣*x*⎦ = 1, ⎣*y*⎦ = 2, (*x* − ⎣*x*⎦) − (*y* − ⎣y⎦) > 0, is obtained. This rule generates a set of clock regions. Above their maximum values, the relations between the clocks are not examined.

Since infinitely many time transitions between a pair or regions are possible, a region succession graph is created.

### 2.1. Timed Automaton-Syntax

Here, we give the definition of timed automata, a base of Uppaal extension. A timed automaton *TA* is a tuple (*L*, *l*_0_, *Z*, *Q*, *E*, *J_l_*) where:*L =* {*l*_0_, *l*_1_, …} is a finite set of *locations*,*l*_0_ ∈ *L* is an *initial location*,*Z* = {*c*_0_, *c*_1_, …} is the set of *clocks*,*Q* denotes a set of *labels* (interpreted as actions on transitions, do not confuse with IMDS actions),every location *l* ∈ *L* is mapped by *J_l_*(*l*) to a set of valuations of clocks in *Z*, over a Cartesian product of ℝ≥0Z, for example, *J_l_*(*l*) = {*c*_1_-*c*_2_ > 2},*E* ⊆ *L* × *Q* × *J_ll_* × 2*^Z^* × *L—*set of *transitions*: *e* = (*l*, *q*, *J_ll_*(*l,l′*), *r*, *l′*) ∈ *E**J_ll_* is a set of functions for pairs *l,l′* ∈ *L*, every transition (*l,l′*)*,* is mapped by *J_ll_*(*l,l′*) to a set of valuations of clocks in *Z* over a Cartesian product of ℝ≥0Z, just as *J_l_*(*l*) for a location *l*,*r* ∈ 2*^Z^* indicates a subset of clocks in *Z* that are reset on transition.

### 2.2. Timed Automaton-Semantics

The semantics of a TA is:

Let (*L*, *l_0_*, *Z*, *Q*, *E*, *J_l_*) be an *TA*.

The semantics is defined as an *LTS*—labeled transition system ⟨*Vertices, vertex_0_,* ▶⟩, where:*Vertices* ⊆ *L* × ℝ≥0Z is the set of LTS vertices for the automaton *TA* (because the term *state* is reserved for IMDS node states, we use the term *vertex* instead), *vertex_0_* = (*l_0_*, *u_0_*) ∈ *Vertices* is the initial vertex, *u_0_* maps all clocks *c* ∈ *Z* to 0.▶ = ▶_t_ ∪ ▶_p_ is the *transition relation* such that:(*l, u*) ▶_t_ (*l, u* + d) if ∀*d′*: 0 ≤ d*′* ≤ d ⇒ *u* + d*′* ∈ *J_l_*(*l*) – timed transition,(*l, u*) ▶_p_ (*l′, u′*) if there exists *e* = (*l, q, J_ll_(l,l′), r, l′*) ∈ *E*

Such that *u* ∈ *J_ll_*(*l*,*l′*); *u′* = [*r* ↳ 0]*u* – progress transition;

where

for d′ ∈ ℝ≥0Z, *u +* d′ maps each clock *c* in *Z* to the value *u*(*c*) + d′,

[*r* ↳ 0]*u*, *r* ⊂ *Z*, denotes the clock valuation, which maps each clock in *r* to 0 and agrees with *u* over *Z*\*r*.

### 2.3. Network of TA

*NTA* is a network of timed automata over a common set of clocks and labels (actions). *NTA* is itself a TA, it is constructed in the following way:

*NTA* = (*L*, *l^0^*, *Z*, *Q*, *E*, *J_l_*) consists of *n TA*, *TA_i_* = (*L_i_, l_i_^0^, Z_i_, Q_i_, E_i_, J_li_*) with *i* = 1, …, *n*. The individual elements of *NTA* are:A set of *locations* is a Cartesian product *L* = *L_1_* × … × *L_n_*, *l* ∈ *L* is a *location vector l* = (*l*_1_, …, *l_n_*), *l_i_* ∈ *L_i_*.*l^0^* ∈ *L* is an *initial location vector l^0^* = (*l_1_^0^*, …, *l_n_^0^*), *l_i_^0^* ∈ *L_i_*.*Z* is a common set of *clocks*—a union of sets of clocks *Z* = *Z*_1_ ∪ … ∪ *Z_n_*.*Q* is a common set of *labels*—symbols on transitions *Q* = *Q_1_* ∪ … ∪ *Q_n_*.*Location invariant functions* are composed into a common function over location vectors *J_l_*(*l*) = *J_l_*_1_(*l*_1_) ∧ … *∧ J_ln_*(*l_n_*).

### 2.4. The Semantics of the Network of TA

Let *TA_i_* = (*L*, *l^0^*, *Z*, *Q*, *E*, *J_l_*).

Let *l^0^* = (*l_1_^0^, …, l_n_^0^*) be the initial location vector.

*l*[*l_i_*/*l_i_*′] denotes the location vector where *l_i_*′ replaces the *i^th^* element *l_i_* of *L*.

The semantics of a network on *n* TA is defined as an *LTS* ⟨*Vertices, vertex_0_,* ▶⟩, where:*Vertices* = (*L*_1_ × … × *L_n_*) × ℝ≥0Z is the set of global LTS vertices,*vertex_0_* = (*l_0_*; *u_0_*) ∈*Vertices* is the initial vertex, *u_0_* maps all clocks *c* ∈ *Z* to 0,▶ = ▶_t_ ∪ ▶_p_ is the *transition relation* defined by:
-(*l*, *u*) ▶_t_ (*l*, *u* + d) if ∀_d′_ 0 ≤ d′ ≤ d ⇒ *u +* d′ ∈ *J_l_*(*l*)—timed transition,-if there exists a symbol *q* ∈ *Q*, for which there exist transitions *e_i_, e_j_, …* in *TA_i_, TA_j_, …* (at least one, but if more, then in distinct automata): *e_i_* = (*l_1i_*, *q*, *J_lli_*(*l_1i_*,*l_2i_*), *r_i_*, *l_2i_*) ∈ *E_i_*, *e_j_* = (*l_1j_*, *q*, *J_llj_*(*l_1j_*,*l_2j_*), *r_j_*, *l_2j_*) ∈ *E_j_*, …then there exists the transition *e* = (*l_1_*, *q*, *J_ll_*(*l_1_*,*l_2_*), *r*, *l_2_*) ∈ ▶_p_ such that*u* ∈ *J_ll_*(*l*[*l_1i_*/*l_2i_*, *l_1j_*/*l_2j_, …*]), *u′* = [*r_i_* ∪ *r_j_* ∪ …↳ 0]*u*—progress transition, Comment: There is a synchronization transition if more than one transition in distinct TA have the same *q* label on their transitions.

### 2.5. The Uppaal Extesion to TA

Transitions are executed simultaneously by the original TA if they are triggered by shared symbols (which enable transitions). A common symbol can synchronize more than two automata. However, standard TA are useless, because they do not apply urgent channels (a timed transition can be executed while a progress transition is enabled) and the do not use variables; thus, no value can be passed between automata that explode the automata layout. Therefore, we give the definition of Uppal timed automata (UTA [13]) in Section 4.

Communication channels replace the common symbols that trigger transitions in Uppaal timed automata. Sending and receiving a signal synchronously via channels is accomplished by utilizing a shared channel symbol and two characters indicating sending and receiving, ! and ?, respectively, as in *chan*! and *chan*? As it specifies the communication direction, this method better reflects distributed systems. Broadcast communication entails multiple automata receiving a signal given by one automaton.

The channel can be labeled *urgent*, in which case transitions are enabled, with common channel symbols enabling these transitions: *chan*! and *chan*?, taking precedence over time flow in locations, preventing communication over an urgent channel from being delayed. However, in IMDS, using such a technique eliminates communication asynchrony. As a result, asynchronous communication requires the introduction of asynchronous channels.

## 3. Integrated Model of Distributed Systems (IMDS)

The most straightforward description of IMDS is as follows: there are distributed *nodes* in the system, which are characterized by their current *states* and *agents* conducting distributed calculations, the progress of which is determined by their current *messages* directed to individual nodes. If the message *matches* the node state, an *action* is *fired* (invoked) that consumes the message and state and provides the node’s *next state* and the agent’s *next message*. Therefore, actions are the relationship between pairs (*message, state*), input and output ones; that is all. However, a more detailed definition should be introduced to define processes in the system and verify the system.

An IMDS system [2] consists of a set of nodes, each represented by its current state as a pair: (node, value). Each node offers services that distinguish different calls to the same node. The set of agents represents distributed computing. In the context of agents, messages are created that invoke node services. The message is a triple: (*agent, node, service*). States and messages are, together, called *elements*. They change their roles as process carriers and means of communication in dual views of distributed systems.

The *action*refers to how the service is carried out (on the node designated by the message). The action switches the node’s current state to the next one and sends the next agent message. This new message might be sent to the same or a different node. Distributed computation is carried out as a series of actions that alter the states of the nodes involved. A relation mapping the state and message to the next state and message can be viewed as an *action* (performed by an agent on a node). The current condition of the node may or may not allow the service to be run (if the status and message *match* or not). The appropriate action is *prepared* if the service can be performed. Pending messages are those that are waiting at a node. At the same node, many actions might be prepared. The action to be performed (to be *fired*) is chosen non-deterministically.

A single action is always performed on a specific node, and the actions of different nodes are performed using *interleaving* [24]. The choice of nodes that will perform the next action is non-deterministic.

The agent *terminates* in this scenario, because a special action does not deliver the next message. As a result, the number of agents may decrease during distributed computing. It is assumed that the system starts with all nodes’ initial states and all agents’ initial messages.

The *configuration* is a ‘snapshot’ of the system; it is a set of states (one state for each node) and messages (maximum one state for each agent, except for terminated agents). The *initial configuration* contains the states of all nodes and messages of all agents. The *input configuration* of the action includes its *input state* and *input message*. The *output configuration* of the action contains the *initial state* and the *initial message* (or only the initial state for the agent-terminating action).

IMDS semantics is defined by the *Labeled Transition System* (LTS [38,39]), in which nodes are configurations and transitions are actions (creating a relation of configuration succession, deduced from actions). The LTS contains all executions of the modeled system.

Sequences of actions (in the system) are called *processes*. The sequence of actions performed by the same node is called the *node process*, and the sequence of actions that performs messages of the same agent is called the *agent process*. Due to the possible non-determinism of the set of actions (more than one action can be defined for a given pair (*message, state*)), the process is actually a graph.

Each system (the entire computation) can be decomposed into *node processes* (which make up the node view) or *agent processes* (which make up the agent view).

Deadlock and termination are expressed in terms of states, messages, and actions [2]. First, consider a node process. The following situations may occur:the current state matches some pending messages in the node—at least one action is ready to be executed; the node process is *running*;the current state does not match any pending messages in the node (or no messages are pending on the node)—a matching message may arrive in the node in the future; the node process is *waiting*;no messages are pending in the node, and none will appear in the future; the node process is *idle*;there are pending messages in the node, but the present state does not match any of them, and there may not be any matching messages in the node in the future; the node process has reached a *deadlock*.

Consider an agent process, which can reach the following cases:
the agent’s message is pending in the node and matches the current state of this node—the action is ready to fire; the agent *runs*;the agent’s message is pending in the node and does not match the current state of this node, but a matching state may occur in the future; the agent is *waiting*;the agent’s message is pending in the node, and neither the current state of this node nor such a condition may occur in the future; the agent is *deadlocked*;the agent process has *terminated,* since no agent’s messages are pending in any node.

### 3.1. Basic IMDS Definition

The IMDS model is based on two sets and a binary relation on the Cartesian product of these two sets. The sets are

*P* = {*p*_1_*, p*_2_*, …, p_Np_*}—finite set of *states* (of *nodes*)*M* = {*m*_1_*, m*_2_*, …, m_Nm_*}—finite set of *messages* (of *agents*)

while the action relation *Λ* is a binary relation on *M* × *P*:*Λ* ⊂ (*M* × *P*) × (*M* × *P*)—set of *actions*

For the elements of *Λ* we use the prefix notation *λ* = ((*m,p*), (*m′,p′*)) ∈ *Λ*. The idea of the IMDS action is presented graphically in Figure 1. The elements of the action, in this and consecutive figures, are given as small blue circles with numbers. There are the references to those elements in the text, for example (1).

The agent’s message invokes an action on the node (the calculation step in a distributed environment) that can be performed in specific states of the node. Performing the action ‘consumes’ the message and state, and creates the next node state and the next agent message.

In the action *λ* = ((*m,p*), (*m′,p′*)) we say that:the pair (*m,p*) *match* (1,31);(*m,p*) is the input pair, (*m′,p′*) is the output pair (2,32);the state *p* is *current*, the message *m* is *pending*;*p′* is the next state, *m′* is the next message.

The definition must be supplemented by the initial sets of states and messages:*P_ini_* ⊂ *P* – set of initial states*M_ini_* ⊂ *M* – states of initial messages.

It should be noted that IMDS is not an automata-based model, although it can be represented as a collection of automata, in various ways; the paper [40] formally presents the conversion of IMDS to node automata or to agent automata. In the present article, we use some graphical notation informally, with node states as vertices and actions as transitions, because such notation is natural and makes it easier to express certain features of the IMDS. However, it should be remembered that such automata are only one of the possible interpretations; there are also others, such as the agent automata mentioned above, as well as pairs of synchronous automata (*custodians* and *messengers* [41]), or the non-automata interpretation as Petri nets shown in [21]. There is also an imperative language Rybu, compatible with IMDS, used by the student to verify their synchronization solutions [42]. Now, we are working on an even higher level language for web service composition. Please note that IMDS is only a set of actions over the quadruple Cartesian product (*M* × *P*) × (*M* × *P*). For programming purposes, these actions can be grouped on nodes, agents, or otherwise, which does not change the essence of the definition of formalism itself.

### 3.2. IMDS System Behavior

The behavior of the IMDS system is represented by labeled transition systems (LTS [38,39]), i.e., a rooted labeled directed graph.

LTS *vertices* are configurations that are sets of current states and pending messages. The *root* is the *initial configuration*, consisting of agents’ *initial messages* and nodes’ *initial states*. States and messages together are called *items*. LTS transitions are defined by actions, transforming their *input configurations* into *output configurations*.

*H* = *P* ∪ *M*—set of *items**T_ini_* = *P_ini_* ∪ *M_ini_*—initial configuration*T* ⊆ *H*—configuration∀*_λ_**_∈_**_Λ_ λ* = ((*m,p*),(*m′,p′*)) *T_inp_*(*λ*) ⊃ {*m,p*}, *T_out_*(*λ*) = *T_inp_*(*λ*)\{*m,p*} ∪ {*m′,p′*}—obtaining *T_out_*(*λ*) from *T_inp_*(*λ*) for an action (31,1)→(32,2)*LTS* = ⟨*N,N_0_,W*⟩, where*N* is a set of *vertices* (configurations {*T*_0_*, T*_1_*, …*}, *T_ini_ = T*_0_);*N*_0_ = *T_ini_* is the *root*;*W* is the set of directed labeled *transitions*, *W* ⊆ *N* × *Λ* × *N*, *W* = {(*T_inp_*(*λ*),*λ**_i_*,*T_out_*(*λ*) | *λ**_i_*∈ *Λ*, *i* = 1, …, ord(*Λ*)}.

The interleaving semantics of the system is assumed [43], i.e., exactly one action is executed at a time.

### 3.3. IMDS Processes

Processes are defined in the IMDS as action sequences. Intermediate elements link the actions in the sequence. If the intermediate elements are the states of a given node, then this is the process of that node, and the messages serve as a means of communication between processes. Conversely, if the messages of a given agent combine the actions of a process, then this is the process of this agent, and the states of the nodes are used to communicate processes. To do this, certain attributes must be assigned to the elements. Therefore, the elements are not atomic, as in the previous chapter. We redefine elements as tuples in four basic sets: *nodes*, *agents*, *values*, and *services*.

The formal definition of IMDS for the purpose of processes extraction is as follows:*S* = {*s*_1_*, s*_2_*, …, s_Ns_*}—finite set of *nodes**A* = {*a*_1_*, a*_2_*, …, s_Na_*}—finite set of *agents**V* = {*v*_1_*, v*_2_*, …, v_Nv_*}—finite set of *values**R* = {*r*_1_*, r*_2_*, …, r_Nr_*}—finite set of *services*

The nodes’ states are defined as pairs (*node, value*): *p* = (*s,v*), *s* ∈ *S*, *v* ∈ *V*, and the messages as triples (*agent, node, service*): *m* = (*a,s,r*), *a* ∈ *S*, *s* ∈ *S*, *r* ∈ *R*. In such a formulation, a message is an invocation of a node’s service by an agent. An action is an execution of a service on a node in the context of an agent.

*P* ⊂ *S* × *V*—set of states,*M* ⊂ *A* × *S* × *R*—set of messages,*H* = *P* ∪ *M*—set of items,Initial sets *P_ini_* and *M_ini_*, set of items *H,* configurations *T* and *T_ini_* are defined over *P* and *M,* as before.

In order to terminate the agent, we add a new type of action, having a pair (message, state) on the input, but only a singleton (state) on the output. The definition of the action relation in terms of agents, nodes, values, and services is as follows:*Λ* ⊂ (*M × P*) *×* (*M × P*) ∪ (*M × P*) *×* (*P*) | (*m,p*)*Λ*(*m′,p′*) ∨ (*m,p*)*Λ*(*p′*), *m* = (*a,s,r*) ∈ *M*, *p* = (*s*_1_*,v*_1_) ∈ *P*, *m′* = (*a_2_,s_2_,r_2_*) ∈ *M*, *p′* = (*s_3_,v_3_*) ∈ *P*, *s*_1_ = *s*, *s*_3_ = *s*, *a*_2_ = *a*

Thus, *Λ* is not strictly a relation, because it contains both quadruples and triples.

We define functions of *m* and *p,* appointing their node and agent: for *m* = (*a,s,r*), Ms(*m*) = *s*, Ma (*m*) =*a*, for *p* = (*s,v*), Ps(*p*) = *s*.

The previous definitions of *T*, *T_inp_*, *T_out_*, and *LTS* hold. The initial configuration contains exactly one state for every node and exactly one message for every agent: initial states of every node and initial message for every agent. For *T_ini_*, ∀*m*_1_*,m*_2_ ∈ *T_ini_*, *m*_1_ ≠ *m*_2_: Ma(*m*_1_) ≠ Ma(*m*_2_); ∀*p*_1_*,p*_2_ ∈ *T_ini_*, *p*_1_ ≠ *p*_2_: Ps(*p*_1_) ≠ Ps(*p*_2_).

The definitions of node and agent processes are as follow:*B*(*s*) = {*λ* ∈ *Λ* | *λ* = (((*a,s,r*),(*s,v*)), ((*a,s′,r′*),(*s,v′*))) ∨ *λ* = (((*a,s,r*),(*s,v*)), ((*s,v′*)))*, s′* ∈ *S, a* ∈ *A, v,v′* ∈ *V, r,r′* ∈ *R*}—node process of the node *s* ∈ *S*,*C*(*a*) = {*λ* ∈ *Λ* | *λ* = (((*a,s,r*),(*s,v*)), ((*a,s′,r′*),(*s,v′*))) ∨ *λ* = (((*a,s,r*),(*s,v*)), ((*s,v′*)))*, s,s′* ∈ *S, v,v′* ∈ *V, r,r′* ∈ *R*}—agent process of the agent *a* ∈ *A*,

The system *views* are

**B** = {*B*(*s*_1_),*B*(*s*_2_),…,*B*(*s_Ns_*) | *s_i_* ∈ *S*}—the *node view* (decomposition of the system to node processes),**C** = {*C*(*a*_1_),*C*(*a*_2_),…,*C*(*a_Na_*) | *a_i_* ∈ *A*}—the *agent view* (decomposition of the system to agent processes).

### 3.4. Automated Deadlock and Termination Identification in IMDS

For model checking, atomic Boolean formulas must be assigned to any configuration in the LTS:*D_s_*—*true* in all configurations, where at least one message is pending at the node *s*,*E_s_*—*true* in all configurations, where at least one action is prepared at the node *s*.*D_a_*—*true* in all configurations, where a message of the agent *a* is pending,*E_a_*—*true* in all configurations, where the action is prepared with a message of the agent *a*,*F_a_*—*true* in all configurations, where a terminating action (with a message of the agent *a* on input) is prepared.

Model checking formulas—CTL version [6]:communication deadlock in node *s*: **EF AG**(*D_s_* ∧ ¬*E_s_*)—a configuration is reachable in which a message is pending at the node *s,* but from this configuration on, no action will be prepared on node *s*;node *s* idle: **AF AG**(¬*D_s_*)—there is a configuration after which no message will arrive at *s*;resource deadlock in the agent *a*: **EF AG**(*D_a_* ∧ ¬*E_a_*)—a configuration is reachable in which a message of the agent *a* is pending but from this configuration on, the message will not match any state;termination of agent *a*: **AF**(*F_a_*)—a terminating action of agent *a* is inevitable.

It should be noted that the above deadlock detection formulas are defined for individual processes (nodes or agents). They allow the finding of *partial deadlocks* (also referred to as local deadlocks in the literature) concerning a limited number of processes, not just total deadlocks (also known as global deadlocks). The total node deadlock can be found in IMDS by means of the formula **EF AG**(∀*s* ∈ *S*: *D_s_* ∧ ¬*E_s_*), and the total agent deadlock by means of the equivalent formula over agents. Typical static deadlock detection methods only detect total deadlock [2]. There are methods for detecting partial deadlocks, but at the cost of limiting the allowed process shape or explicitly specifying a deadlock using model-specific formulas. Finding a partial deadlock using general formulas is an original achievement in IMDS.

The above IMDS definition supports all of the features of the distributed system:*communication duality*: Any system can be broken down into node processes that communicate via messages (the output state of the action is the carrier of the node process, while the output message is the means of communication) or agent processes communicating through the states of the nodes (the output message is the carrier agent process, while the output state is the means of communication);*locality*: Each action on a node is dependent on the node’s current state, and one of a set of pending messages on that node; no incident from outside (except messages received by the node) can affect the behavior of the node;*autonomy*: each node independently decides which of the defined actions can be performed in the current situation; in other words, the nodes decide for themselves if, and when, the messages are accepted and what actions they will cause;*asynchrony*: the node receives a message when it is ready for it; otherwise the message is pending; there are no synchronous operations in the model, such as simultaneous transitions on shared symbols in Büchi automata [3] or timed automata [18], and no joint operations of nodes or agents; synchronous sending and receiving operations in CSP [44], Occam [45], or Uppaal Timed Automata [14], synchronous operations on the complementary input and output ports in CCS [44]; node and agent autonomy is implemented using asynchronous operations: sending a message to a node or setting a new node state for subsequent agent operations are the only ways to influence the behavior of nodes and agents;*asynchronous channels*: communication between nodes is one-way (communication in the opposite direction has its separate channel) and may appear synchronous because the message appears on the receiving node immediately after it is sent; however, asynchrony is modeled by the possibility of deferring the message’s acceptance; the message can wait a long time, even forever, before being accepted;*automated verification*: the four above-mentioned temporal formulas are used to locate communication deadlocks in the processes of individual nodes, idleness of nodes, deadlocks over resources in agent processes, and agent termination, regardless of the structure of the verified system; therefore, they form the basis of the design of the automatic Dedan verifier, which can be used without knowledge of time logic and model checking.

We will present the translation of timed IMDS to timed automata for two reasons. First, TA is the most commonly used formalism on concurrent systems; thus, the translation of T-IMDS to TA gives a formal definition of its semantics. Second, the Dedan model checker is based on explicit state space representation, allowing for verification of small and medium systems. Large systems, in which more than ten nodes with complicated structures are contained, are exported from Dedan to Uppaal model checker for their verification.

## 4. Timed IMDS (T-IMDS)

In the timed version of IMDS, all of the mentioned functions are preserved: the duality of communication, locality, autonomy, asynchrony, and automated verification. In addition, time restrictions, which can last over time, are imposed on elements of the distributed system:*time durations* of actions (fixed or range),*time delays* of inter-node channels (fixed or range).

This collection of time constraints is smaller than the possibilities offered by TA; for example, time invariants of locations implementing staying in states, and differences of clock values are not included in T-IMDS. However, the selected time-related functions are best addressed to modeling distributed systems: differences in clock values (between distributed nodes) assume some knowledge about the global state. Limiting the time spent in locations implementing node states cannot be mixed with operations on urgent channels that are needed for the implementation of asynchronous channels (see later in this chapter); this is a limitation of the UTA syntax.

The behavior of a distributed system is determined by its LTS. All possible sequences of actions are included in the LTS. Time constraints are intended to exclude certain behaviors, due to violations of time restrictions imposed on actions and channels. Such a modification of the system behavior may, for example, prevent a deadlock (exclude the sequence leading to a deadlock) or cause a deadlock (for example, a process that is intended to meet a given condition may become stuck prematurely due to time constraints).

### 4.1. Syntax

The syntax of timeless IMDS is simply a set of actions of the form (((*a*,*s*,*r*),(*s*,*v*)),((*a*,*s*′,*r*′),(*s*,*v*′))), plus agent-terminating actions (((*a*,*s*,*r*),(*s*,*v*)),((*s*,*v*′))), denoted {a.s.r, s.v} -> {a.s′.r′, s.v′} and {a.s.r, s.v} -> {s.v′}. To simplify the description, in the description of timed formalism we will omit the agent terminating actions. An additional syntax is used to define types of nodes and agents, their parameterization, declaration of node and agent variables, and their initialization.

In notation, the dispersion of the duration of the action, in the form of a range with a lower and upper bound, is inserted between the input and output items of the action, for example a duration range (*z1,z2*) for an action *λ* has the form: *λ* = ((*m_inp_,p_inp_*),(*m,p*)) = (((*a,s,r*),(*s,v*)),(*z1,z2*)((*a,s_out_,r_out_*),(*s,v_out_*))). A range can have open or closed bounds: (*z1,z2*), <*z1,z2*), (*z1,z2*>, <*z1,z2*>. The notation of ranges is taken from T-IMDS, where round parentheses denote open bounds and angle brackets denote closed bounds. The timed action has the form {a.s.r, s.v} -> (z1,z2){a.s’.r’, s.v’} in T-IMDS source format, and angle brackets can be applied for closed bounds.

Channel delays are defined as ranges in **channels** {...} phrase, for all channels: (*d1,d2*), for all channels leading to a given node *s*, or all elements of a vector *s*[…] of nodes: –>*s*(*d1,d2*), or between individual nodes: *s1*–>*s2*(*d1,d2*). In addition, individual elements of node vectors can be used: –>*s*[2](*d1,d2*), *s1*[1]–>*s2*[3](*d1,d2*).

An example of a system consisting of two distributed semaphores and two nodes using those semaphores, specified in T-IMDS in the input form of the Dedan verifier, is presented Listing 1 (the nodes are called *servers* in the IMDS language).

Listing 1.

 1.**system** two_semaphores;

 2. **server**: sem(**agents** a1,a2;**servers** sa1,sa2),

 3. **services** {wait, signal},

 4. **states** {up, down},

 5. **actions** {

 6. {a1.sem.wait, sem.up} → (2,3>{a1.sa1.ok_wait, sem.down} 

 7. {a1.sem.signal, sem.down} → (2,3>{a1.sa1.ok_signal, sem.up}

 8. {a2.sem.wait, sem.up} → (2,3>{a2.sa2.ok_wait, sem.down} 

 9. {a2.sem.signal, sem.down} → (2,3>{a2.sa2.ok_signal, sem.up}

10. };

11. **server**: proc(**agents** Ag;**servers** sem[2]),

12. **services** {start, ok_w, ok_s},

13. **states** {initial, first, second, end},

14. **actions** {

15. {Ag.proc.start, proc.initial} -> <0>{Ag.sem[1].wait, proc.first},

16. {Ag.proc.ok_w, proc.first} -> <0>{Ag.sem[2].wait, proc.second},

17. {Ag.proc.ok_w, proc.second} -> <0>{Ag.sem[1].signal, proc.first},

18. {Ag.proc.ok_s, proc.first} -> <0>{Ag.sem[2].signal, proc.second},

19. {Ag.proc.ok_s, proc.second} -> <0>{proc.end},

20. };

21. **servers** semaphore[2]:sem,process[2]:proc;

22. **agents** Ag[2]; 

23. **channels** {<0>};

24. **init** -> {

25.  process[1](Ag[1],semaphore[1,2]).initial,

26.  process[2](Ag[2],semaphore[2,1]).initial,

27. <j=1..2> semaphore[j](Ag[1..2],process[1..2]).up,

28. <j=1..2> Ag[j].process[j].start,

29. }.

This is the specification in the node view. It is simply a collection of actions on particular nodes. A set of node type specifications defines a system (enclosed by **server** …};—lines 2–10, 11–20), node and agent instance (variable) declarations (**agents** …, **servers** …—lines 21,22), and an initial configuration phrase (**init** → lines 24–29). A node type heading contains a set of formal parameters, such as the agents and nodes employed in the node type’s activities. Formal parameters, such as *Ag*[2] and *sem*[2], can be vectors (line 11). A set of services (lines 3,12), a set of states (lines 4,13), and a set of actions are allocated to each node (lines 6–9, 15–19). An action *λ* = (((*a,s,r*),(*s,v*)), ((*a,s_out_,r_out_*),(*s,v_out_*))) has the form {a.s.r, s.v} → (x,y){a.sout.rout, s.vout}. The time bounds (x,y) limit the action duration. Services and states can be vectors (not in this example, see the source code of AVGS system in Section 7.2). Repeaters may precede actions in a node type definition for its compactness (for example, in an AVGS system Section 7.2, 2 repeaters are used for some actions). Vectors can be used to arrange node and agent instances (lines 21,22). The **channels** phrase (line 23) specifies channel delays, in this example, the channels have no delay. The delays can be assigned to each channel individually. Actual parameters are bound to formal parameters in the initialization part (line 24). The nodes’ initial states and the agents’ initial messages are also assigned.

### 4.2. Semantics

The execution of the action is divided into three phases for states and four phases for messages, as shown in Figure 2. The complete environment of the action *λ*, with the actions delivering the input items *m_inp_* and *p_inp_*, is presented in Figure 3. Of course, more than one action can deliver an input item of the action. Due to non-determinism, the action can deliver its output items, *m* and *p*, to multiple consecutive actions. The phases of the action are as follows:The current input state *p_inp_* (1) and pending input message *m_inp_* (31) match; therefore, they can invoke the action *λ* = ((*m_inp_,p_inp_*),(*m,p*)): (31,1)→(32d,2c). If multiple actions are enabled in a node, the choice is nondeterministic. The first phase (31,1)→(32a,2a) is a *reception* of the message *m_inp_* and *invocation* of the action.Time *duration* of the action begins, which lasts between t_λ min_(*λ*) and t_λ max_(*λ*). Counting the time duration is the second phase (32a,2a)→(32b,2b).When the time duration ends, the new pair of (*m,p*) is *generated* (32b,2b)→(32c,2c). From this moment, the state *p* is available for invoking the next action in the node. The message *m* is *sent* to the target node, and it must be propagated to become accessible.The last phase is message delivery, in which the channel delay between t_ch min_(*ch*) and t_ch max_(*ch*) is counted (32c)→(32d). After the *delay*, the message *m* becomes available for invocation of the next agent action, in the target node with its current state.

The semantics of T-IMDS (timed LTS) is:For every action *λ* = ((*m_inp_,p_inp_*),(*m,p*)) (3), the range bounds of action duration t*_λ_*
_min_(*λ*) and t_λ max_(*λ*) are defined.The set of channels *CH* of the form *ch* = (*a*,*s_inp_*→*s*) are defined: ∀*λ* = ((*m_inp_,p_inp_*),(*m,p*)): ∃ *ch*: *ch* = (Ma(*m_inp_*), Ms(*m_inp_*)→Ms(*m*)). Totally K channels, indexed 1, …, K. The channel transmitting the output message of the action *λ* is denoted *ch_λ_*.For every *ch*, the range bounds of channel delay t_ch min_(*ch*), t_ch max_(*ch*) are defined.Delay time is defined for a channel between given nodes; therefore, for all agents sending messages along the channel, the range of delay is equal: ∀*a_1_*,*a_2_*: ∀*ch_1_* = (*a_1_*, *s_inp_*→*s*), *ch_2_* = (*a_2_*, *s_inp_*→*s*): t_ch min_(*ch_1_*) = t_ch min_(*ch_2_*) ∧ t_ch max_(*ch_1_*) = t_ch max_(*ch_2_*); however, this does not influence the shape of the LTS.A timed configuration consists of messages, states, their derivatives, node time values *ct_s_*, and channel time values *ct_ch_*: *T_t_* = (*T* = {*m_tai_*, *p_tsj_* | *i* = 1, …, Na, *j* = 1, …, Ns, *m_tai_* ∈ *M_t_*, *p_tsi_* ∈ *P_t_*}, *ct_s1_, …, ct_sNs_*, *ct_ch1_, …, ct_chK_*), where Na is the number of agents and Ns is the number of nodes. *T* is called a set of *items*.The set of states *p* (1,2b) and derivatives *p_λt_* (2a), *p_λ_* (2b)*_e_*: *P_t_* = {*p_i_*, *p_iλjt_*, *p_iλje_* | *I* = 1, …, card(*P*), *j* = 1, …, card(*λ*), *λ**_j_* = ((*m_inp_,p_inp_*),(*m,p_i_*)) ∈ *Λ*, *p_iλjt_* = (*p_i_*,*λ**_j_*), *p_iλje_* = (*p_i_*,*λ**_j_*)}The set of messages *m* (31,32d) and derivatives *m_λt_* (32a), *m_λe_* (32b), *m_chλ_* (32c): *M_t_* = {*m_i_*, *m_iλjt_*, *m_iλje_*, *m_ichλj_* | *i* = 1, …, card(*M*), *j* = 1, …, card(*Λ*), *k* = 1, …, *K*, *λ**_j_* = ((*m_inp_,p_inp_*),(*m_i_,p*)) ∈ *Λ*, *m_iλjt_* = (*m_i_,λ_j_*), *m_iλje_* = (*m_i_,**λ_j_*), *m_ich_**_λj_*= (*m_i_,ch_λj_,λ_j_*) }.For *λ* = ((*m_inp_,p_inp_*),(*m,p*)) we define that Ps(*p_λt_*) = Ps(*p_λe_*) = Ps(*p*), Ms(*m_λt_*) = Ms(*m_λe_*) = Ms(*m_chλ_*) = Ms(*m*), Ma(*m_λt_*) = Ma(*m_λe_*) = Ma(*m_chλ_*) = Ma(*m*).Each *p_λt_* has two attributes: t_λ min_(*p_λt_*) and t_λ max_(*p_λt_*).Each *m_chλ_* has two attributes: t_ch min_(*m_chλ_*) = t_ch min_(ch*_λ_*) and t_ch max_(*m_chλ_*) = t_ch max_(ch*_λ_*).The root vertex in the LTS is the initial timed configuration is *T_0t_* = (*T_0_* = {*m_0a1_*, …, *m_0aNa_*, *p_0s1_, …, p_0sNs_* | *m_0ai_*∈*M_0_*, *p_0si_* ∈ *P_0_*}, 0, …, 0, 0, …, 0).

The vertices in the LTS can contain three types of pair and a singleton causing the progress transitions (0-time transitions):12.(*m_inp_,p_inp_*) (31,1) is the input pair of an action *λ*,13.(*m_inpλt_,p_λt_*) (32a,2a) for a transition ending the time duration of the action *λ*,14.(*m_λe_,p_λe_*) (32b,2b) for generation of the output pair (*m,p*) of the action *λ*,15.(*m_chλ_*) (32c) for a transition ending the time delay of the message *m* generated in the action *λ*.

In addition, we have a pair and a singleton causing the timed transitions:16.(*m_inpλt_,p_λt_*) (32a,2a) for a timed transition modeling sub-periods of time in the action duration of the action *λ*,17.(*m_chλ_*) (32c) for a timed transition modeling sub-periods of time in the time delay of the message *m* generated in the action *λ*;

In execution of the action *λ*, the following pairs occur in some configurations in the LTS: (*m_inp_,p_inp_*), (*m_inpλt_,p_λt_*), (*m_λe_,p_λe_*), (*m_chλ_,p*). Finally, *m* is not related to *p* or any derivative of *p*, because it takes some time for *m* to become operational, after channel delay. However, it can occur in a pair with *p* if the node does not begin execution of any action during the message *m* passing through the channel.

The transitions in the LTS are

18.reception/invocation transition (31,2)→(32a,2a)—for *T_t_*: (∃{*m_inp_,p_inp_*} ⊂ *T*: ∃*λ* ∈ *Λ*: *λ* = ((*m_inp_,p_inp_*),(*m,p*))) ⇒ *T’_t_* = (*T*\{*m_inp_,p_inp_*} ∪ {*m_inpλt_,p_λt_*}, previous ct_s1_,…,ct_sNs_ except *c_Ps(p inp)_* := 0, previous *ct_ch1_,…,ct_chK_*);19.generation/send transition (of a new *m* and *p*) (32b,2b)→(32c,2c)—for *T_t_*: (∃{*m_λe_,p_λe_*} ⊂ *T*: *λ* = ((*m_inp_,p_inp_*),(*m,p*))) ⇒ *T’_t_* = (*T*\{*m_λe_,p_λe_*} ∪ {*m_chλ_,p*}, previous ct_s1_, …, ct_sNs_, previous ct_ch1_, …, ct_chK_ except *c_ch_**_λ_* := 0);20.two transitions—action duration timed transition (32b,2b)→(32b,2b) and duration end timeless transition (32b,2b)→(32c,2c)—for *T_t_*: ¬(∃ {*m_inp_,p_inp_*} ⊂ *T*: ∃*λ* ∈ *Λ*: *λ* = ((*m_inp_,p_inp_*),(*m,p*))) ∧ //no action to invoke ¬(∃ {*m_λe_,p_λe_*} ⊂ *T*: *λ* = ((*m_inp_,p_inp_*),(*m,p*))) ∧ // no message to generate (∃ {*m_inpλt_,p_λt_*} ⊂ *T*) | *s*=Ps(*p**_λt_*) ⇒ (t_λ min_(*λ*) < *ct_s_* < t_λ max_(*λ*) ⇒ *T’_t_* = (*T*\{*m_inpλt_,p_λt_*} ∪ {*m_λe_,p_λe_*}, previous ct_s1_,…,ct_sNs_ except *c_s_* := 0, previous ct_ch1_,…,ct_chK_), //action duration ended (*ct_s_*<t_λ max_(*λ*) ⇒ *T″_t_* = (*T*, *c_si_* := previous ct_si_ + δ, *i* = 1..Ns, *c_chj_* := previous ct_chj_ + δ, *j* = 1..K));21.two transitions—channel delay timed transition (32c)→(32c) and delay end timeless transition (32c)→(32d)—for *T_t_*: ¬(∃ {*m_inp_,p_inp_*} ⊂ *T*: ∃*λ* ∈ *Λ*: *λ* = ((*m_inp_,p_inp_*),(*m,p*))) ∧ //no action to start ¬(∃ {*m_λe_,p_λe_*} ⊂ *T*: *λ* = ((*m_inp_,p_inp_*),(*m,p*))) ∧ // no message to generate (∃ *m_chλ_*∈*T* ) ⇒ (t_ch min_(*ch_λ_*) < ct_ch_ < t_ch max_(*ch**_λ_*) ⇒ *T’_t_* = (*T*\{*m_chλ_*} ∪ {*m*}, previous ct_s1_, …, ct_sNs_, previous ct_ch1_,…,ct_chK_ except *c_ch_**_λ_* := 0)), //channel delay ended (*ct_ch_* < t_ch max_(*m_chλ_*) ⇒ *T″_t_* = (*T*, *c_si_*:= previous ct_si_ + δ, *i* = 1..Ns, *c_chj_* := previous ct_chj_ + δ, *j* = 1..K)).22.If multiple transitions come out of an LTS vertex, the choice is nondeterministic. However, if both reception/generation transition and duration/delay end transition are possible in the current configuration, the latter is not included into LTS.23.The general limit for δ in the parallel timed transitions is that for all ct_si_ in *T_t_*, *λ**_1_* = ((*m_1inp_,p_1inp_*),(*m_1_,p_1_*)), Ps(*p_1inp_*) = *s_i_*, *p_λ1t_* ∈ *T*, and all ct_ch__λ2_ in *T_t_*, *λ**_2_* = ((*m_2inp_,p_2inp_*),(*m_2_,p_2_*)), *m_ch_*_λ*2*_ ∈*T*, δ < min(t*_λ_*
_max_(*p_λ1t_*)–ct_si_, t_ch max_(*ch*_λ*2*_)–ct_ch__λ2_). In every inequality t_min_ < ct, ct < t_max_, the relation should be replaced by ≤ if the corresponding range bound is closed. If all t_λ max_(*p_λ1t_*) and all t_ch max_(*m_chλ2_*) in *T_t_* are for closed upper bounds of time ranges, then the relation in the inequality < for δ should be replaced by ≤.

The latter condition says that we must not exceed the upper bound of any remaining action duration/channel delay.

## 5. Uppaal Timed Automata

### 5.1. The Syntax of Uppaal TA

A collection of timed automata and real-valued clocks are commonly used to define parallel systems. We expand the concept to include variables that can enable transitions and be allocated to transitions to conform with Uppaal TA.

An Uppaal timed automaton *UTA* is a tuple (*L*, *l_0_*, *Z*, *CH*, *Q*, *E*, *J_l_, O, Ō_0_*) where

*L=* {*l_0_, l_1_, …*} is a finite set of *locations*,*l_0_* ∈ *L* is an *initial location*,*Z* = {*c_0_, c_1_,* …} is the set of *clocks*,*CH* is a set of symbols called *channels*,*Q* denotes a set of *labels* (interpreted as actions on transitions, do not confuse with IMDS actions), they represent *send* and *receive* operations on a channel: *ch*!, *ch*?, *ch* ∈ *CH*; outside the automaton, internal labels are ignored and replaced by *τ*,every location *l* ∈ *L* is mapped by *J_l_*(*l*) to a set of valuations of clocks in *Z*, over a Cartesian product of ℝ≥0Z, for example, *J_l_*(*l*) = {*c_1_*-*c_2_* > 2}; Restriction. As in verification tools, e.g., Uppaal [14], we limit location invariants to downwards closed constraints of the form: *x ≤ n* or *x* < *n* where *n* is a natural number,*O* = {*o*_1_, *o*_2_*,* …} finite set of *variables*, for which we define:*V_i_* = {v_i1_, v_i2_, …}—finite, integral set of values of variable *oi* ∈ *O*,*P_V_* = *V*_1_ × *V*_2_ × … × *V*_*ord*(*O*)_—a Cartesian product of values of variables *o*_1_, *o*_2_, …, *o*_*ord*(*O*)_ ∈ *O*,*Ō* = (v_1_, v_2_, …, v_ord(O)_); v_i_ ∈ *V_i_*; *Ō* ∈ *P_V_*—*vector* of values of variables in *O*,*Ō_0_* = (v_1_^0^, v_2_^0^, …, v_ord(O)_^0^)—*initial vector* of values of variables in *O*,*E* ⊆ *L* × *Q* × *J_ll_* × 2*^Z^* × 2*^P^**^V^*× *F* × *L*—set of *transitions*: *e* = (*l*, *q*, *J_ll_*(*l,l′*), *r*, *b*, *f*, *l′*) ∈ *E*
*J_ll_* is a set of functions for pairs *l,l’*∈*L*, every transition (*l,l’*) is mapped by *J_ll_*(*l,l′*) to a set of valuations of clocks in *Z* over a Cartesian product of ℝ≥0Z, just as *J_l_*(*l*) for a location *l*,*r* ∈ 2*^Z^* indicates a subset of clocks in *Z* that are reset on transition,*b* ∈ 2*^P^**^V^*—set of vectors of variable values enabling a transition; Comment: typically *b* is presented as equalities and inequalities between variables in *O* and constants, connected by Boolean operators, for example (*o*_1_ < 3) ∧ (*o*_2_ ≥ 6),*F* is a set of functions on variables in *O*, *f* ∈ *F, f*: *P_V_* → *P_V_*—unction assigning new values to the variables in *O*; *f/i* restricts this function to the value of variable *o_i_*. Comment: In Uppaal TA, the function is given as a set of assignments *o_i_* = expression over variables in *O* and integer constants (all other variables are left unchanged).

### 5.2. The Semantics of UTA

The Semantics of a UTA is:

Let (*L*, *l_0_*, *Z*, *CH*, *Q*, *E*, *J_l_, O, Ō_0_*) be an *UTA*.

The semantics is defined as an *LTS*—Labeled Transition System ⟨*Vertices, vertex_0_,* ▶⟩, where:*Vertices* ⊆ *L* × *P_V_* × ℝ≥0Z is the set of LTS vertices for the automaton *UTA* (because the term *state* is reserved for IMDS node states, we use the term *vertex* instead),*vertex_0_* = (*l_0_*, *Ō_0_*, *u_0_*) ∈ *Vertices* is the initial vertex, *u_0_* maps all clocks *c* ∈ *Z* to 0.▶ = ▶_t_ ∪ ▶_p_ is the *transition relation* such that:(*l, Ō, u*) ▶_t_ (*l, Ō, u*+d) if ∀*d’*: 0 ≤ d′ ≤ d ⇒ *u* + d′ ∈ *J_l_*(*l*)—timed transition,(*l, Ō, u*) ▶_p_ (*l*′*, Ō*′*, u*′) if there exists *e* = (*l, q, J_ll_(l,l*′*), r, b, f, l*′) ∈ *E* such that *u* ∈ *J_ll_*(*l*,*l*′); *u*′ = [*r* ↳ 0]*u* and *Ō* ∈ *b* and *Ō*′ = *F*(*Ō*)—progress transition;

where

for d’ ∈ ℝ≥0Z, *u +* d′ maps each clock *c* in *Z* to the value *u*(*c*) + d′,

[*r* ↳ 0]*u*, *r* ⊂ *Z*, denotes the clock valuation, which maps each clock in *r* to 0 and agrees with *u* over *Z*\*r*.

### 5.3. Network of UTA

*NUTA* is a network of Uppaal timed automata over a common set of clocks and labels (actions), a common set of variables, and a common initial vector of their values. *NUTA* is itself a UTA, it is constructed in the following way:

*NUTA* = (*L*, *l^0^*, *Z*, *CH*, *Q*, *E*, *J_l_, O, Ō^0^*) consists of *n UTA*, *UTA_i_* = (*L_i_, l_i_^0^, Z_i_, CH_i_, Q_i_, E_i_, J_li_, O_i_, Ō_i_^0^*) with *i* = 1, …, *n*. The individual elements of *NUTA* are:A set of *locations* is a Cartesian product *L* = *L*_1_ × … × *L_n_*, *l* ∈ *L* is a *location vector l* = (*l*_1_, …, *l_n_*), *l_i_* ∈ *L_i_*.*l^0^* ∈ *L* is an *initial location vector l^0^* = (*l_1_^0^*, …, *l_n_^0^*), *l_i_^0^* ∈ *L_i_*.*Z* is a common set of *clocks*—a union of sets of clocks *Z* = *Z*_1_ ∪ … ∪ *Z_n_*.*CH* is a common set of channels—a union of sets of channels *CH* = *CH*_1_ ∪ … ∪ *CH_n_*, it can be ignored because the labels in *Q* disappear in the construction of *NUTA*.*Q* is a common set of *labels*—symbols on transitions: the labels on channels (*ch*!, *ch*?) disappear on the construction on *NUTA* (they are replaced by *τ*) according to the semantic rules given below.*Location invariant functions* are composed into a common function over location vectors *J_l_*(*l*) = *J_l_*_1_(*l*_1_) ∧ … ∧ *J_ln_*(*l_n_*).*O* is a common *set of variables* (union of sets of variables *O*_1_ ∪ … ∪ *O_n_*), *Ō*—vector of their values, *P_V_*—the Cartesian product of sets of values of all variables in *O*, *Ō*_0_—a common vector of their *initial values*.

The *NUTA* graph is constructed in such a way that transitions between the compound locations of *NUTA* (starting from initial compound location *l*^0^) are chosen as common timed transitions, interleaved progress transitions having labels, and common progress transitions of pairs of *UTA* with matching *ch*! and *ch*? labels, interleaved with the transitions of all other *UTA*. Additionally, the set of transitions in *NUTA* can be restricted by conjunctions of time invariants *J_l_* in timed transitions, and by conjunctions of functions *J_lli_* and intersections of *b_i_*. Last, all those rules met by the semantics of *NUTA* are expressed as the *LTS* of a set *UTA* below.

### 5.4. The Semantics of the Network of UTA

Let *UTA_i_* = (*L*, *l^0^*, *Z*, *CH*, *Q*, *E*, *J_l_, O, Ō^0^*).

Let *l^0^* = (*l_1_^0^, …, l_n_^0^*) be the initial location vector.

*l*[*l_i_*/*l_i_*′] denotes the location vector where *l_i_*′ replaces the *i^th^* element *l_i_* of *L*.

The semantics of a network on *n* UTA is defined as an *LTS* ⟨*Vertices, vertex_0_,* ▶⟩, where:*Vertices* = (*L*_1_ × …× *L_n_*) × *P_V_* × ℝ≥0Z is the set of global LTS vertices,*vertex_0_* = (*l_0_*; *Ō_0_*; *u_0_*) ∈ *Vertices* is the initial vertex, *u*_0_ maps all clocks *c* ∈ *Z* to 0,▶ = ▶_t_ ∪ ▶_p_ ∪ ▶_!?_ is the *transition relation* defined by:3a.(*l*, *Ō*, *u*) ▶_t_ (*l*, *Ō*, *u* + d) if ∀_d′_ 0 ≤ d’ ≤ d ⇒ *u +* d′ ∈ *J_l_*(*l*)—timed transition,3b.(*l*, *Ō*, *u*) ▶_p_ (*l*[*l_i_*/*l_i_′*], *Ō′*, *u′*); if there exists (*l_i_*, *τ*; *J_lli_*(*l_i_*,*l_i_′*), *r_i_*, *b_i_, f_i_, l_i_′*) ∈ *E_i_* such that *u* ∈ *J_lli_*(*l*[*l_i_*/*l_i_′*]); *u′* = [*r_i_* ↳ 0]*u* and *Ō* ∈ *b_i_* and *Ō′* = *f_i_*(*Ō*)—progress transition,3c.(*l*, *Ō, u*) ▶_!?_ (*l*[*l_j_*/*l_j_′*, *l_i_*/*l_i_′*], *Ō′, u′*) if there exist two transitions (*l_i_*, *ch*?, *J_lli_*(*l_i_*,*l_i_′*), *r_i_*, *b_i_, f_i_, l_i_′*) ∈ *E_i_* and (*l_j_*, *ch*!, *J_llj_*(*l_j_*,*l_j_′*), *r_j_*, *b_j_, f_j_, l_j_′*) ∈ *E_j_* such that *u* ∈ *J_ll_*(*l*[*l_j_*/*l_j_′*, *l_i_*/*l_i_′*]), *u′* = [*r_i_* ∪ *r_j_* ↳ 0]*u* and *Ō* ∈ *b_i_* and *Ō* ∈ *b_j_*—synchronization transition (a special kind of progress transition), new values of variables in *O* are calculated:for every *o_k_* ∈ *O*, *f_i_*(*o*_k_) = v_k_ and *f_j_*(ok) = v_k_′ or *f_j_*(*o*_k_) = v_k_ and *f_i_*(*o_k_*) = v_k_′ Comment: At least one of the functions *f_i_*, *f_j_* must be an identity function for a given variable that returns the same value as its argument (it is disregarded); the other must be in effect (it gives the variable’s new value). This criterion prohibits incoherent assignments to the same variable in the automata *UTA_j_* and *UTA_k_*; it is met by the construction in the translation of T-IMDS to UTA, as an assignment is applied in only one of the pair’s automata;If both ▶_t_ and ▶_!?_ are possible from given vertex (*l*; *Ō*; *u*), then ▶_t_ is not inserted into *LTS*,If both ▶_p_ and ▶_!?_ are possible from given vertex (*l*; *Ō*; *u*), then ▶_p_ is not inserted into *LTS*; Comment: In this way, Uppaal urgent channels are achieved; only such channels are applied in the translation of T-IMDS to UTA,If two transitions are possible, the choice is non-deterministic (however, both are inserted into LTS because it defines all possible behaviors).

## 6. Translation of T-IMDS to Timed Automata

### 6.1. Example

To explain the translation rules, let us concentrate on the actions of the *sem* node from the example in Section 5.1, with actions duration (2,3>). It implements *wait* and *signal* services. Two agents, *a1* and *a2*, use the semaphore. Each agent comes from its private node: *sa1* for *a1,* and *sa2* for *a2*. The time constraints are given in Listing 2.

Listing 2.

 5. **actions** {

 6. {a1.sem.wait, sem.up} → (2,3>{a1.sa1.ok_wait, sem.down}   //λ_1_

 7. {a1.sem.signal, sem.down} → (2,3>{a1.sa1.ok_signal, sem.up} //λ_2_

 8. {a2.sem.wait, sem.up} → (2,3>{a2.sa2.ok_wait, sem.down}   //λ_3_

 9. {a2.sem.signal, sem.down} → (2,3>{a2.sa2.ok_signal, sem.up} //λ_4_

10. }

For every node *s*, an Uppaal timed automaton *s* is declared. In fact, an UTA type is defined just as a node type can be defined in T-IMDS, but this is a syntactic manipulation for the programmer’s convenience. The UTA variable is equipped with its own clock *c_s_* and a set of locations and derivatives, in the example *c_sem_*, *up, up**_λ_**_2t_, up**_λ_**_2e_, up**_λ_**_4t_, up**_λ_**_4e_, down, down**_λ_**_1t_, down**_λ_**_1e_, down**_λ_**_3t_, down**_λ_**_3e_*. For storing the service name invoked by messages, variables *a1_sem* and *a2_sem* are applied for the agents in *sem*. For nodes *sa1* and *sa2*, variables *a1_sa1* and *a2_sa2* are applied. The set of values for *a1_sem* and *a2_sem* variables is {none, sem_wait, sem_signal}, for *a1_sa1* {none, sa1_ok_wait, sa1_ok_signal}, and *a2_sa2* {none, sa2_ok_wait, sa2_ok_signal}. The node clocks are *c_sem_*, *c_sa1_*, and *c_sa2_*.

The initial state is chosen outside the node type, in the *init* section, to allow different instances to have different initial states; *up* or *down* in our example. Assume that the node *sem* has the initial state *up*. The set of UTA locations and transitions for automaton *sem* is as follows (for comparison in Boolean expressions over variables we use a double equals operator, which follows the Uppaal rule; for assignment and clock reset, we use equals character preceded by a colon (:=) to clearly distinguish it from comparison and to agree with Uppaal):(31,1) location *down*: initial location—no, time invariant—no (4),-(31,1)→(32a,2a) transition: next location—*up**_λ_**_2t_*, condition—*a1_sem*==sem_signal, assignments—no, time constraint—no, clock reset—*c_sem_*:=0, channel synchronization: *ch_a1_sem*? (4)→(8)→(5)-(31,1)→(32a,2a) transition: next location—*up**_λ4t_*, condition—*a2_sem*==sem_signal, assignments—no, time constraint—no, clock reset—*c_sem_*:=0, channel synchronization: *ch_a2_sem*? (4)→(8)→(5)(32a,2a) location *up**_λ_**_2t_*: initial location—no, time invariant—*c_sem_* ≤ 3 (5),-(32a,2a)→(32b,2b) transition: next location—*up**_λ_**_2e_*, condition—no, assignments—*a1_sem*:=none, *a1_sa1*:=sa1_ok_up, time constraint—2 < *c_sem_* ≤ 3, clock reset—*c_sem_*:=0, channel synchronization—no (5)→(9)→(6)(32b,2b) location *up**_λ_**_2e_*: initial location—no, time invariant—no (6),-(32b,2b)→(32c,2c) transition: next location—*up*; condition—no, assignments—no, time constraint—no, clock reset—*c_sem_*:=0, channel synchronization: *ch_a1_sem_sa1*! (6)→(10)→(7)(32a,2a) location *up**_λ4t_*: initial location—no, time invariant—*c_sem_* ≤ 3 (5),-(32a,2a)→(32b,2b) transition: next location—*up**_λ4e_*, condition—no, assignments—*a2_sem*:=none, *a2_sa2*:=sa2_ok_up, time constraint—2 < *c_sem_* ≤ 3, clock reset—*c_sem_*:=0, channel synchronization—no (5)→(9)→(6)(32b,2b) location *up**_λ4e_*: initial location—no, time invariant—no,-(32b,2b)→(32c,2c) transition: next location—*up*; condition—no, assignments—no, time constraint—no, clock reset—*c_sem_*:=0, channel synchronization: *ch_a2_sem_sa2*! (6)→(10)→(7)(2c) location *up*: initial location—no, time invariant—no (7),…etc.

### 6.2. Translation Rules

Since the channels leading from different nodes to the node *s* are combined together, we use the notation (*a*,→*s*) for the channel passing messages of the agent *a* to the node *s*. The rules for translation of *λ* = ((*m_inp_,p_inp_*),(*m,p*)), *m_inp_* = (*a,s,r*), *m* = (*a,s_out_,r_out_*) are depicted in Figure 4, and message *m* transfer along the channel *ch* = (*a*,→*s_out_*) from T-IMDS to UTA in Figure 5. Three channel versions are shown: (a) the basic form, (b) the 0-time channel without delay, and (c) the channel receiving messages of the agent coming from various nodes with different delays. Please do not mistake the *wait* location of a channel with the wait service offered by semaphores in the example. We numbered all the elements of T-IMDS and its UTA implementation to be referred to in the following translation rules:*sever s* ⇒ timed node automaton *s* with clock *c_s_* (reset on every transition, used to count actions duration),*state p* of automaton *s* (1,2c) and derivatives (*p**_λ_**_t_, p**_λe_*) (2a,2b) ⇒ locations *p, p**_λ_**_t_, p**_λ_**_e_* in timed node automaton *s* (4,5,6,7),*message m* (32,32d) and derivatives (*m**_λ_**_t_, m**_λ_**_e_, m_ch_**_λ_*) (32a,32b,32c) ⇒ pairs of message variable *a_s* values and locations of automaton *s* and automaton *ch*, details below,*agent a* → set of variables (for every node visited by the agent) {*a_s_1_, a_s_2_, …*} (8),*channel ch* ⇒ automaton *ch* (Figure 3) with clock *c_ch_* (reset on every transition, used to count channel delay),*message m_inp_* pending at the node *s* (1) ⇒ *a_s* == s_r (8), all other *a_s_x_* == none, *s_x_* ≠ *s*, channel *ch* in location *send* (11,18,24), variable *a_s* has a set of values {none, s_r, s_r_1_, s_r_2_, …} (8) (services *r*, *r*_1_, *r*_2_ distinguish between messages sent to the node *s* by the agent *a*),*derivative item p_λt_* for action *λ* (duration running) (2a) ⇒ location *p_λt_* (5),*derivative item p_λe_* for action *λ* (duration ended) (2b) ⇒ location *p_λe_* (6),*message m_inp_* (stable message *m_inp_*) (32) ⇒ pair (*send,a_s*==s_r); (8) location *send* in channel (*a*,→*s*) (11,18,24),*derivative item m_inp_**_λ_**_t_* for action *λ* ⇒ pair (*p**_λ_**_t_*, *a_s* == s_r) (5,8,14/19/29); note that *a_s_out_*==none,*derivative item m**_λ_**_e_* for action *λ* ⇒ pair (*p**_λ_**_e_*, *a_s_out_* == s_out__r_out_) (6,9); note that *a_s* == none,*derivative item m_ch_**_λ_* for action *λ* ⇒ location *wait* of the channel *ch* automaton (15/20/25/26); note than *a_s* == none, *a_s_out_* == s_out__r_out_ (9),*configuration T* ⇒ set of locations of node automata representing node states and derivatives and values of pairs (location of a node automaton *s_x_*/channel automaton *ch_z_*, value of *a_y__s_x_*) representing messages and derivatives,*actual value of action duration in node s/channel ch delay* in *T_t_*—the value of the clock *c_s_*/clock *c_ch_* (9/27/28),*duration range lower bound* t_λ min_(λ) ⇒ lower bound of time constraint of the transition *p*_λ*t*_→*p*_λ*e*_: t_λ min_(λ) < *c_s_* (9),*duration range upper bound* t_λ max_(λ) ⇒ upper bound of time constraint of the transition *p*_λ*t*_→*p*_λ*e*_: *c_s_* < t_λ max_(λ) (9) and of time invariant of location *p*_λ*t*_: *c_s_* < t_λ max_(λ) (5),*delay range lower bound* t_ch min_(*ch*) ⇒ lower bound of time constraint of the transition *wait*→*send* of the channel automaton, t_ch min_(*ch*) < *c_ch_* (16/27/28),*delay range upper bound* t_ch max_(*ch*) ⇒ upper bound of time constraint of the transition *wait*→*send* of the channel automaton, *c_ch_* < t_ch max_(*ch*) (16/27/28) and of time invariant of *wait* location: *c_ch_* < t_λ max_(*ch*) (13/21/22),*action λ* (3), (31,1)→(32a,2a)→(32b,2c)→(32c,2c), (32c)→(32d) ⇒ sequence: message and state(*m_inp_,p_inp_*)-receive transition(*m_inp_,p_inp_*)→(*m_inp_**_λt_*,*p**_λt_*)-duration location(*p**_λt_*)-timed transitions(action duration in *p**_λt_*)-end of duration transition(*m_inp_**_λt_*,*p**_λt_*)→(*m**_λe_*,*p**_λe_*)-end of duration location(*p**_λe_*)-send transition(*m**_λe_,p**_λe_*)→(*m_ch_**_λ_*,*p*), (11/18/24,4) → (14/19/29,8) → (12/17/23,5) → (9) → (12/17/23,6) → (15/20/25/26,10) → (13/18/21/22,7),*receive transition* (*m_inp_,p_inp_*)→(*m_inp_**_λt_*,*p**_λt_*) (31,1)→(32a,2a) ⇒ channel automaton *ch_as_* = (*a*,→*s*) in location *send*, node *s* automaton in location *p_inp_*, urgent channel synchronization (*ch_a_s*! on (*m_inp_,p_inp_*)→(*m_inp_**_λt_*,*p**_λt_*), *ch_a_s*? on *send*→*idle*), condition *a_s* == s_r fulfilled, on transition *c_s_* is reset to 0, (11/18/24,4) → (14/19/29,8) → (12/17/23,5),*end of action duration transition* (*m_inp_**_λt_*,*p**_λt_*)→(*m**_λe_*,*p**_λe_*) (32a,2a)→(32b,2b) ⇒ node *s* automaton in location *p**_λt_*, clock *c_s_* value t_λ min_(*λ*) < *c_s_* < t_λ max_(*λ*), on transition *a_s* := none, *a_s_out_* == s_out__r_out_, *c_s_* is reset to 0, (12/17/23,5) → (9) → (12/17/23,6),*generation of next m,p and send transition* (*m**_λe_*,*p**_λe_*)→(*p,m_ch_**_λ_*) (32b,2b)→(32c,2c) ⇒ node *s* automaton in location *p**_λe_*, channel *ch* automaton in location *idle*, urgent channel synchronization (*ch_a_s_s_out_*! on (*m**_λe_*,*p**_λe_*)→(*p,m_ch_**_λ_*), *ch_a_s_s_out_*? on *idle*→*wait*), (12/17/23,6) → (15/20/25/26,10) → (13/18/21/22,7),*advancing duration* of an action (32b,2b)→(32b,2b) or *channel delay* (32c)→(32d) ⇒ timed transitions—within time invariants of all locations with time invariant—to a next time region, limited by outgoing transitions time range upper bounds, t_λ max_ for action durations in *p**_λt_* locations (5–9), and t_ch max_ for channel delays in *wait* locations (13–16/21–27/22–28),*asynchronous channel* between nodes *ch* (32c)→(32d) ⇒ an asynchronous channel is compound of two synchronous urgent channels (Figure 5); asynchronous channel is inactive in the *idle* location (12/17/23); the signal that the message is ready is obtained from the sending automaton via input urgent channel *ch_a_s_out_*: urgent channel synchronization (*ch_a_s_out_*!, *ch_a_s_out_*?) (15/20/25/26); then, the time delay is counted in the *wait* location (13/21/22); it lasts between t_ch min_ and t_ch max_, as the transition ending the channel delay is followed for clock values *c_ch_* t_ch min_(*ch*) < *c_ch_* < t_ch max_(*ch*) (16/27/28); finally, the signal sending the message to the target automaton via output urgent channel *ch_a_s_s_out_* is issued: urgent channel synchronization (*ch_a_s_s_out_*!, *ch_a_s_s_out_*?) (14/19/27/28); if the receiving *s* automaton is not ready to accept the message—the synchronization on *ch_a_s* is deferred: a message is pending (11/18/24); an own local clock *c_ch_* is used to count the time delay of an asynchronous channel (13–16/21-–27/22–28).

Note that, by construction, exactly one variable *a_s_x_* has a value other than none, because, initially, exactly one variable of the agent *a* has a value representing the service in initial message of *a*, and setting a value ≠ none to a variable *a_s_y_* of target automaton *s_y_* resets the value of *a_s_x_* to none (9). Moreover, only the channel to the node *s_x_* appointed by the initial message of the agent *a* (*a*,→*s_x_*) is in the *send* location (11/18/24), all other channels of the agent *a* are in their *idle* locations (12/17/23).

### 6.3. Translation of the Example

Having the translation rules, which define the semantics of T-IMDS by construction, let us show how the fragment of the *sem* node (in the example presented in Section 5.1) is translated to UTA. We choose the two transitions, threaded by the input and output states, and we show the messages incoming to and outgoing from the node. Let us assume a channel delay (1,2) between sem and sa1 in both directions. The time constraints are given in Listing 3.

Listing 3.

 6. {a1.sem.wait, sem.up} → (2,3>{a1.sa1.ok_wait, sem.down} //λ_1_

 7. {a1.sem.signal, sem.down} → (2,3>{a1.sa1.ok_signal, sem.up} //λ_2_

23. **channels** {(1,2)};

The image of the two actions is presented in Figure 6, and the UTA implementation in Figure 7.

### 6.4. Equivalence between T-IMDS and UTA

We argue that the LTS of the T-IMDS system is exactly the same as the LTS of the set of UTA implementing the system. The proof is rather extensive because numerous elements and cases must be analyzed; therefore, we give the proof in the Appendix B. Here we present some equivalences between the T-IMDS system and UTA implementation using the rules presented in Section 6.2. We use references to the enumeration numbers in Section 4.2 (T-IMDS semantics), 5.4 (UTA semantics), and 6.2 (translation rules).

The configuration *T_t_* (states and derivatives, messages and derivatives, the current time region determining abstraction class of node clocks and channel clocks) entirely defines the situation in T-IMDS. In UTA, locations of node and channel automata, values of variables (there are only *a_s* variables), and current time region define the situation (we do not use the term ‘state’ for unambiguity) (4.2.5, 5.4.1, 6.2.2, 6.2.9, 6.2.10, 6.2.11, 6.2.12). Finally, every set of items in *T_t_* corresponds to a separate set of UTA locations and variable values. Every reachable set of locations and variable values maps to a set of T-IMDS items. Moreover, the region succession graph agrees with advancing the time according to minimum and maximum time bounds (of action duration and channel delay) (4.2.5, 5.4.3a, 6.2.14).The same concerns the initial configuration of T-IMDS and the initial situation in UTA (4.2.11, 5.4.2).Every one of the T-IMDS transitions (*m_inp_,p_inp_*)→(*m_inp_**_λt_*,*p**_λt_*)→(*m**_λe_*,*p**_λe_*)→(*m_ch_**_λ_*,*p*) depends only on items enumerated in pairs, independently of any other item present in *T_t_* (4.2.18–4.2.21, 6.2.20–6.2.22). Only the transition *m_ch_**_λ_*→*m* does not depend on the state of the message issuing node (it can be *p* or some other item if the next action is invoked) (4.2.21, 6.2.23). In UTA, corresponding transitions depend only on the current location and values of variables representing pending messages. The variables are local to the pairs of messages sending and messages receiving automata, and only the sending automaton can change the variable value (4.2.20, 6.2.21). The automaton collaborates with channel automata, and by construction, the input channel is in the synchronizing location *send* (synchronous channel put (!) on a transition outgoing from *send* location) if *m_inp_* is pending (4.2.18, 6.2.20). The output channel is in the *idle* location (with synchronous channel get (?) on a transition outgoing from *idle*) while the action producing the output message is in progress (4.2.21, 6.2.24).Timed transitions in T-IMDS are concurrent for every clock (node clocks and channel clocks) (4.2.20, 4.2.21, 6.2.23). The same is true for UTA (4.4.3a).Timed transition in T-IMDS is possible if no progress transition is enabled (*m_inp_,p_inp_*)→(*m_inp_**_λt_*,*p**_λt_*), (*m**_λe_*,*p**_λe_*)→(*m_ch_**_λ_*,*p*) (4.2.20, 4.2.21). The same concerns duration ending and delay ending transitions (*m_inp_**_λt_*,*p**_λt_*)→(*m**_λe_*,*p**_λe_*), *m**_λe_*→*m* (4.2.20, 4.2.21). In UTA, all channels are urgent, whose effect is the same: precedence of progress transitions (they correspond to T-IMDS progress transitions) over timed transitions, and over transitions outgoing from time-counting locations (corresponding to duration ending and delay ending transitions) (5.4.4, 6.4.23).In both models, the nondeterministic choice is performed if multiple transitions are enabled (4.2.22, 5.4.6).

## 7. Examples

### 7.1. Simple Example—Two Semaphores

Let us consider two agents, starting from their own nodes, using two semaphores, each one on a separate node. The agents use the semaphores crosswise, i.e., agent *a1* calls *wait* on the first, then on the second semaphore, and agent *a2* calls *wait* on the second, then on the first semaphore. The T-IMDS code is presented in Section 4.1. A deadlock is evident because the semaphores are used crosswise (lines 25,26).

In the timed experiment, we assigned the duration ranges <0> to the actions in *sem* type, and (0,1) to the actions in *proc* type. The time constraints are given in Listing 4.

Listing 4.

6. {a1.sem.wait, sem.up} -> <0>{a1.sa1.ok_wait, sem.down}

…

15. {Ag.proc.start, proc.initial} -> (0,1){Ag.sem[1].wait, proc.first},

…

The channels have 0 delay.

The deadlock appears for the timed system, as before. Figure 8 presents an example of the system trace. It is a sequence diagram of the node operation, with global time values on the right (yellow). Other timeless verification examples can be found in [46,47].

If we lengthen the duration of the actions in *proc*[1] node (the action preceded by an index ?1), as below, the deadlock disappears, because the agent *A*[2] manages to perform wait operations on both semaphores before the agent *A*[1] can perform its first wait operation. The time constraint is given in Listing 5.

Listing 5.

15.?1{Ag.proc.start, proc.initial} -> (8,9) {Ag.sem[1].wait, proc.first}

Figure 9 shows a final part of the witness of both agents inevitable termination.

### 7.2. Practical Example: Automated Vehicle Guidance System

#### 7.2.1. Timeless Verification

An automatic vehicle guidance system (AVGS) is an example of a distributed system for verification. The timeless version of the system and its verification is described by Czejdo et al. [48]. The system, depicted in Figure 6, is made up of road markers and parking lots that communicate to guide autonomous moving platforms (AMPs) from *Lot_E1* to *Lot_E2,* and vice versa. In *Marker_M*, an apparent conflict can be resolved by utilizing *Lot_M* in a staggered arrangement. The controllers of lots and markers are represented by six nodes (Figure 10), with a protocol for requesting and granting the road segments maintained by the controllers (Figure 11). Of course, if *controller_2*’s road segment is occupied, the ok message may be delayed. *MarkerM* has a more sophisticated approach, because it allows for overtaking. The system is described from the perspective of communicating controllers in the node view. The AVGS code is shown in IMDS source notation in the node view. Node names are shortened to *L_E*[2], *L_M*, *M_E*[2] and *M_M.* The source code is given in Listing 6.

Listing 6.

1.**system** AVGS;

2.**server:** M_E(**agents** AMP[2];**servers** M_M,L_E), 

3.**services** {tryM[2],tryL,okM[2],okL,takeM,takeL},

4. //M—going from M_M, L—going from L_E, 

5. //try—test access, ok—accept, take—enter

6.**states** {empty,reservedM,reservedL,occupied}, 

7.**actions** {

8.<i=1..2> {AMP[i].M_E.tryL, M_E.empty} -> 

<0>{AMP[i].L_E.ok, M_E.reservedL},

9.<i=1..2> {AMP[i].M_E.takeL, M_E.reservedL} -> 

 <0>{AMP[i].M_M.tryE[i], M_E.occupied},

10.<i=1..2><j=1..2>{AMP[i].M_E.okM[j], M_E.occupied} ->

 (1,10){AMP[i].M_M.takeE[j], M_.empty},

11.<i=1..2><j=1..2>{AMP[i].M_E.tryM[j], M_E.empty} -> 

 <0>{AMP[i].M_M.okE[j], M_E.reservedM},

12.<i=1..2><j=1..2>{AMP[i].M_E.tryM[j], M_E.reservedL} -> 

 <0>{AMP[i].M_M.notE[j], M_E.reservedM},

13.<i=1..2><j=1..2>{AMP[i].M_E.tryM[j], M_E.occupied} -> 

 <0>{AMP[i].M_M.notE[j], M_E.occupied},

14.<i=1..2> {AMP[i].M_E.takeM, M_E.reservedM} -> 

<0>{AMP[i].L_E.try, M_E.occupied},

15.<i=1..2> {AMP[i].M_E.okL, M_E.occupied} -> 

(1,10){AMP[i].L_E.take, M_E.empty},

16.};

17.**server:** M_M(**agents** AMP[2];**servers** M_[2],L_M), 

18.**services** {tryE[2],tryL[2],okE[2],notE[2],okL[2],takeE[2],takeL[2],switch[2]},

19.**states** {empty,reservedE[2],reservedL[2],occupied},

20.**actions** {

21.//going to M_E1 or M_E2 

22.<i=1..2><j=1..2>{AMP[i].M_M.tryE[j], M_M.empty} -> 

<0>{AMP[i].M_E[j].okM[j], M_M.reservedE[j]},

23.<i=1..2><j=1..2>{AMP[i].M_M.takeE[j], M_M.reservedE[j]} -> 

<0>{AMP[i].M_M.switch[3-j], M_M.occupied},

24.<i=1..2><j=1..2>{AMP[i].M_M.switch[j], M_M.occupied} -> 

<0>{AMP[i].M_E[j].tryM[j], M_M.occupied},

25.<i=1..2><j=1..2>{AMP[i].M_M.okE[j], M_M.occupied} -> 

(1,10){AMP[i].M_E[j].takeM, M_M.empty},

26.//on a way to M_E1 or M_E2 may go to L_E if M_Ei is occupied

27.<i=1..2><j=1..2>{AMP[i].M_M.notE[j], M_M.occupied} -> 

<0>{AMP[i].lotM.try[j], M_M.occupied},

28.<i=1..2><j=1..2>{AMP[i].M_M.okL[j], M_M.occupied} -> 

(1,10){AMP[i].L_M.take[j], M_M.empty},

29.// from M_M—going to M_E1 or M_E2

30.<i=1..2><j=1..2>{AMP[i].M_M.tryL[j], M_M.empty} -> 

<0>{AMP[i].L_M.ok[j], M_M.reservedL[j]},

31.<i=1..2><j=1..2>{AMP[i].M_M.takeL[j], M_M.reservedL[j]} -> 

<0>{AMP[i].M_E[j].tryM[j], M_M.occupied},

32.<i=1..2><j=1..2>{AMP[i].M_M.okE[j], M_M.occupied} -> 

(1,10){AMP[i].M_E[j].takeM, M_M.empty},

33.};

34.**server**: L_E(**agents** AMP[2];**servers** M_E), 

35.**services** {start,try,ok,take},

36.**states** {empty,reserved,occupied}, 

37.**actions** {

38.<i=1..2> {AMP[i].L_E.try, L_E.empty} -> 

<0>{AMP[i].M_E.okL, L_E.reserved},

39.<i=1..2> {AMP[i].L_E.take, lotE.reserved} -> 

<0>{L_E.occupied},

40.<i=1..2> {AMP[i].L_E.start, L_E.occupied} -> 

<0>{AMP[i].M_E.tryL, L_E.occupied},

41.<i=1..2> {AMP[i].L_E.ok, L_E.occupied} -> 

(1,10){AMP[i].M_E.takeL, M_E.empty},

42.};

43.**server**: L_M(**agents** AMP[2];**servers** M_M),

44.**services** {try[2],ok[2],take[2]},

45.**states** {empty,reserved[2],occupied[2]}, 

46.**actions** {

47.<i=1..N><j=1..2>{AMP[i].L_M.try[j], L_M.empty} -> 

 <0>{AMP[i].M_M.okL[j], L_M.reserved[j]},

48.<i=1..N><j=1..2>{AMP[i].L_M.take[j], L_M.reserved[j]} -> 

<0>{AMP[i].M_M.tryL[j], L_M.occupied[j]},

49.<i=1..2><j=1..2>{AMP[i].L_M.ok[j], L_M.occupied[j]} -> 

(1,10){AMP[i].M_M.takeL[j], L_M.empty},

50.};

51.**servers** M_E[2],M_M,L_E[2],L_M;

52.**agents** AMP[2];

53.**channels** {(1,10)};

54.**init** -> { 

55.<j=1..2> M_E[j](AMP[1..2],M_M,lotE[j]).empty,

56. M_M(AMP[1..2],M_E[1,2],L_M).empty,

57.<j=1..2> lotE[j](AMP[1..2],M_E[j]).occupied,

58. L_M(AMP[1..2],M_M).empty,

59.<j=1..2> AMP[j].L_E[j].start,

60.}.

The agent view of the system shows it from the point of view of *AMP* vehicles. The difference is generally in the grouping actions of individual agents rather than of nodes.

The Dedan program (Appendix A) performs the temporal verification. Overtaking in *Marker_M* is successful, but there is a possibility of deadlock when an *AMP* occupies *Marker_E1*, and the other *AMP* tries to drive from *Lot_E1* to *Marker_E1*. A counterexample, showing the deadlock in the node view, representing the cooperation of segment controllers, is presented in Figure 12. The counterexample is represented by a chart that resembles a sequence diagram. The upper part shows the initial states of all nodes on a pink background and the sequences of states and messages that lead to an erroneous situation in a given process (agent names on a green background occur in the agent view). The heading shows the names of the nodes on a pink background (agent names on a green background occur in the agent view). A state identifier is displayed with a light blue background. The agent responsible for entering the specified state is depicted on a dark blue background. When sending a message, the agent identifier is displayed on a light yellow background; while receiving a message, the service name is displayed on a yellow background. The last section displays the final deadlock configuration, including node states and pending messages.

In the timelines of individual nodes, messages from the various agents interleave. The counterexample can be displayed in the agent view, in which timelines of individual agents are shown. This form of counterexample, showing the behavior of individual AMPs, is presented in [48].

#### 7.2.2. Timed Verification

The time of message passing and the time of road segment occupation are undefined in the timeless system. T-IMDS allows observing the behavior with a specified duration of actions and communication delays. Time constraints applied in the T-IMDS variant are:Time of movement between segments (issuing the *take* message) is between 1 and 10.The channel delay is assumed (1,10).

Setting constraints (1,10) yields the ‘nearly no restrictions’ paradigm, which is comparable to a timeless system.

Such time limits being imposed on the AVGS system result in the deadlock presented in Figure 12, as in the timeless system. However, the shorter time constraints (4,5) for taking a segment and (0,1) for channel delay model the simultaneous movement of the two AMPs, which begin practically synchronously from *Lot_E1* and *Lot_E2*. In this case, the deadlock is broken: the AMPs use *Lot_M* to evade in *Marker_M*, but no evasive necessity occurs in *Marker_E1* and *Marker_E2* (time constraints do not allow the AMPs to come into a conflict in edge markers). We present the final fragment of the witness of the modified system, with shortened time constraints, in Figure 13.

## 8. Conclusions

This article presents the IMDS formalism and the Dedan verification environment. They support the modeling of systems using asynchronous communication of autonomous elements, which is natural in distributed environments. The node view and the agent view of the verified system can be observed, which highlight different aspects of distributed system behavior. The node view refers to the client–server model of distributed systems. In contrast, the agent view is similar to the remote procedure call model [5] (however, it is generally unnecessary for the agent process to return to the calling node in IMDS). Deadlocks can be found in both views, showing counterexamples and witnesses from the perspective of communicating nodes or agents traveling between the nodes and changing the nodes’ states.

Universal temporal formulas, unrelated to the structure of a given system, allow finding deadlocks without any knowledge of temporal logic and model checking. The location of deadlocks and distributed termination checking is performed in ‘push the button’ style.

The conversion of IMDS systems to timed automata allows the possibility of verification with real-time constraints. This is suitable for real distributed systems, where communication protocols and time-dependent behavior (such as streaming, monitoring, games) require taking time into account. Both communication delays and the time duration of nodes’ and agents’ actions can be modeled.

Verification of an existing system with the actual values of time constraints can determine if a deadlock is possible. The deadlock reveals, for example, a traffic jam. On the other hand, the threshold values of time constraints (lower and upper bounds) that cause a deadlock can be identified in a series of experiments.

Important cases include embedded systems controlling mechanical or chemical equipment, in which the system behavior depends on the time of real-world phenomena and activities. An example of such a system is presented in this article, in which time delays in controllers’ communication and the time duration of AMPs moving through road segments are modeled. IMDS and Dedan may be used for the ‘rapid prototyping’ of distributed controllers, in which sub-controllers coordinate their behavior by simple protocols; following the internet of things (IoT [49]) paradigm. We have recently started a project of train management, where the balise placement is planned (track equipment), and we estimated the maximum velocity at which the driver is safe in order to notice and acknowledge the messages coming from the balises. All elements of the system (the train, balises, and driver) act asynchronously, with their own timing constraints. Verification can be carried out for various values of time constraints.

The model is not free from limitations. The most important limitation is that distributed computations are performed by the agents, and messages are the carriers of agents. This prohibits communication in broadcast style. Multicast communication can be achieved using a set of ‘sleeping’ agents, which come into operation for multicasting. We used this technique in the verification of BPMN (business process model and notation [50]) processes [51].

The Dedan is based on explicit state–space representation, allowing for the verification of small and medium systems. Large systems are exported from Dedan to Uppaal for their verification, and the semantics of both representations are equal.

The Dedan environment has been effectively employed in the Warsaw University of Technology’s Institute of Computer Science’s operating systems laboratory. The students verify their synchronization solutions during classes.

## Figures and Tables

**Figure 1 sensors-22-01157-f001:**
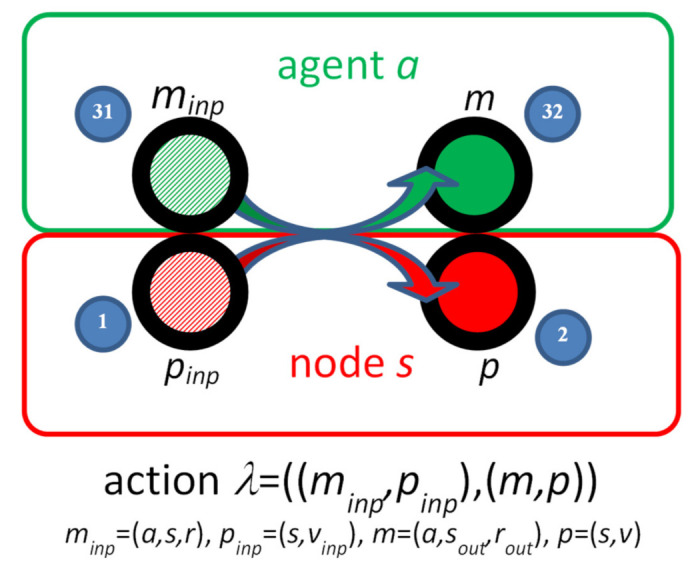
Action in IMDS: relation in (*M* × *P*) × (*M* × *P*).

**Figure 2 sensors-22-01157-f002:**
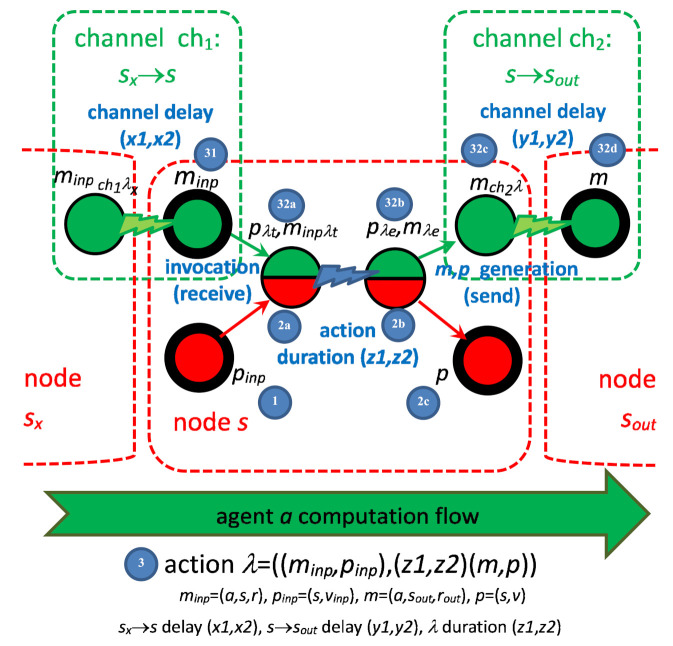
Progress transitions (thin arrows) and timed transitions (thick arrows) in a T-IMDS action occurring in node s and agent a. Elements taking part in the action are surrounded by solid edges, an element of the previous action is surrounded by dashed edges. Every action consists in four steps: message receiving (acceptance and invoking the action), action in progress (time duration), new items generation (message and state), and message delivery (channel time delay).

**Figure 3 sensors-22-01157-f003:**
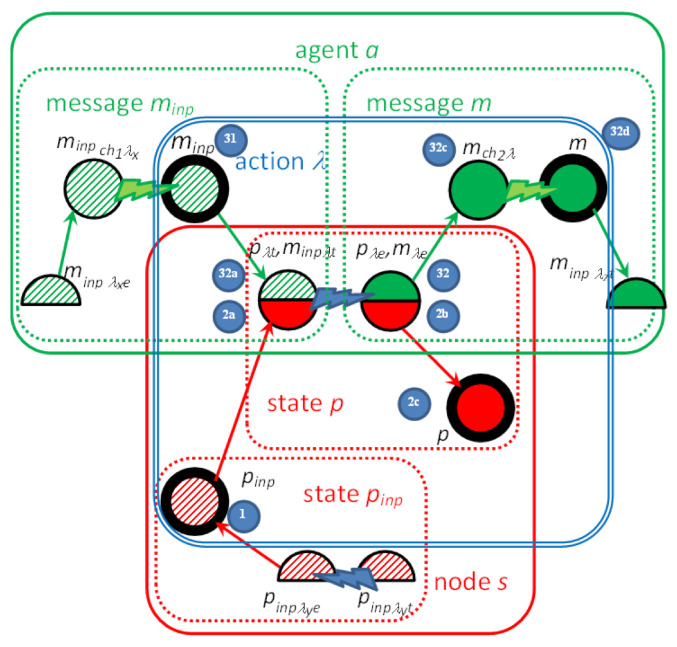
Illustration of the action *λ* = ((*m_inp_,p_inp_*),(*m*,*p*)) with its input elements: message *m_inp_* with derivatives and state *p_inp_* with derivatives. The message *m_inp_* is delivered by the action *λ**_x_* and the state *p_inp_* by the action *λ**_y_*. The input items of the action *λ*, *λ**_x_*, and *λ**_y_*, are also presented with their derivatives. The next action in agent *a* is *λ**_z_*.

**Figure 4 sensors-22-01157-f004:**
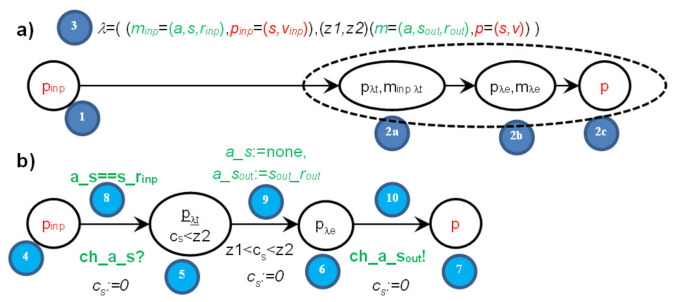
(**a**) action of IMDS with time durations between *z1* and *z2* (**b**) translation to Uppaal TA. For Uppaal TA: underlined font in location—time invariant of location, regular font in location—location name, regular font in transition label—time constraints of transition, *italic font*—update executed on transition, **bold font**—Boolean condition enabling transition or synchronization.

**Figure 5 sensors-22-01157-f005:**
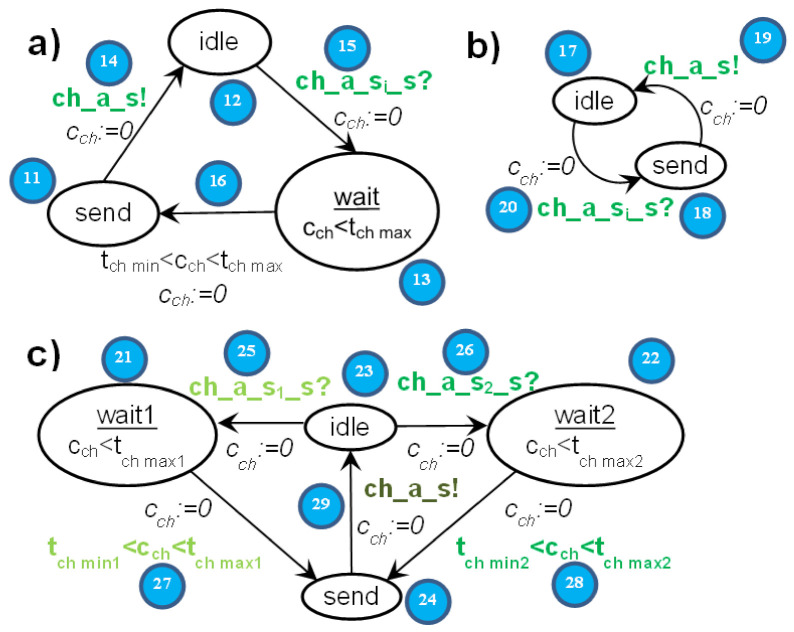
Implementation of communication between nodes in the context of the agent a (asynchronous channel): (**a**) basic asynchronous channel for messages directed to the node s; (**b**) the channel for sending messages from the node s to itself, or other 0-time channels; (**c**) passing messages from multiple nodes (two in this case) to the node s.

**Figure 6 sensors-22-01157-f006:**
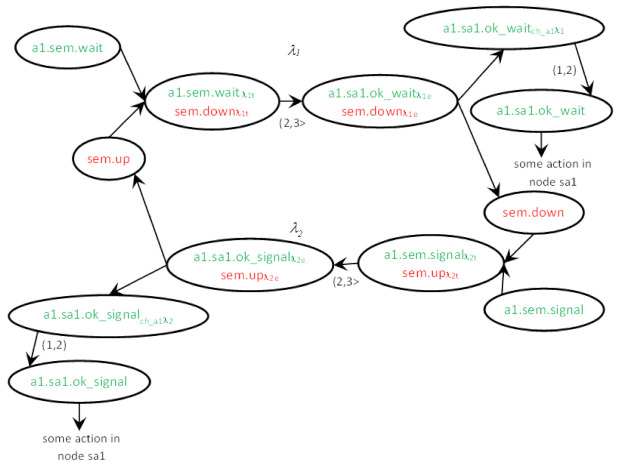
The image of two actions *λ*_1_ = {a1.sem.wait, sem.up} → (2,3>{a1.sa1.ok_wait, sem.down} and *λ*_2_ = {a1.sem.signal, sem.down} → (2,3>{a1.sa1.ok_signal, sem.up}.

**Figure 7 sensors-22-01157-f007:**
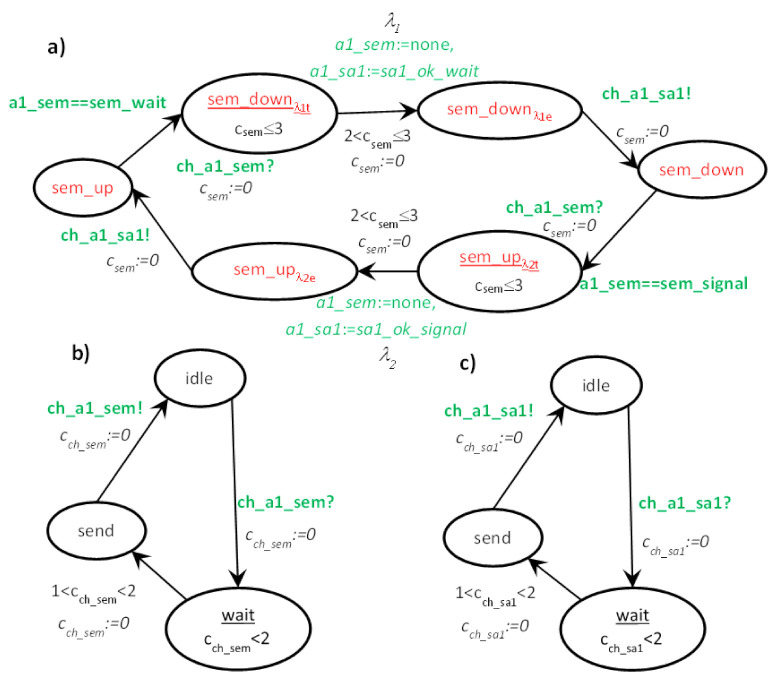
The UTA implementation of the two actions presented in Figure 6, and channels delay (1,2), (**a**) the implementation of the transitions in node automaton *sem*, (**b**) the implementation of the asynchronous channel transferring messages to the node *sem*, (**c**) the implementation of the asynchronous channel transferring messages to the node *sa1*.

**Figure 8 sensors-22-01157-f008:**
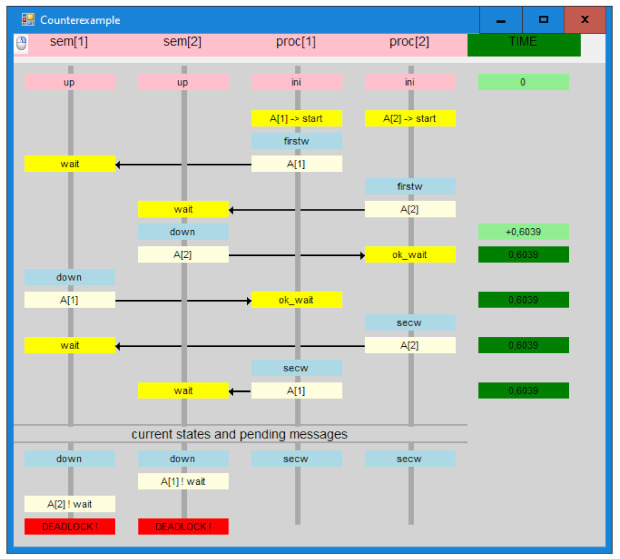
A counterexample showing a deadlock in the timed verification of *two semaphores*.

**Figure 9 sensors-22-01157-f009:**
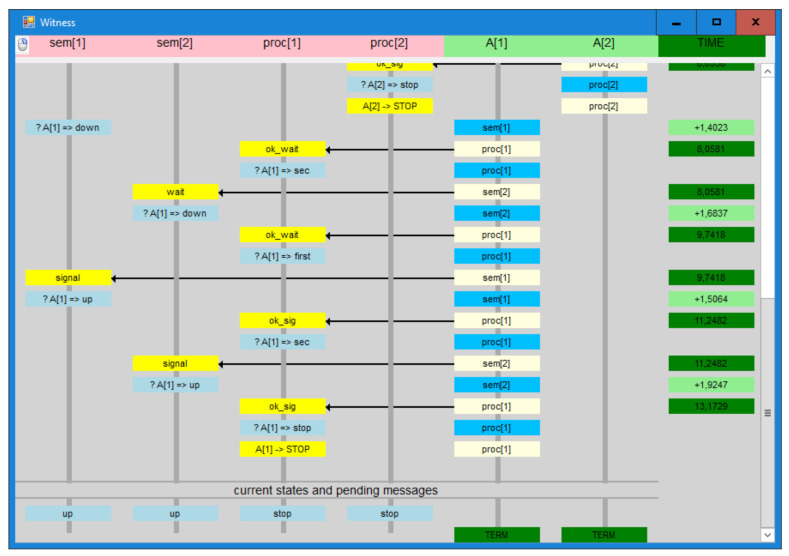
A witness showing a termination of both agents in timed *two semaphores*.

**Figure 10 sensors-22-01157-f010:**
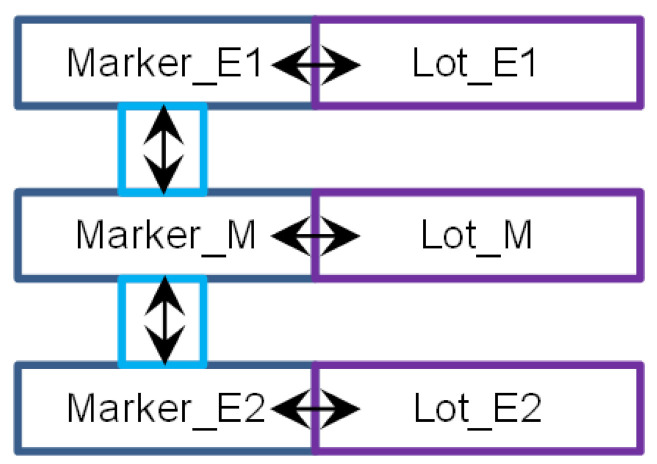
Scheme of road segments (with their markers) and parking lots in an automated vehicle guidance system AVGS.

**Figure 11 sensors-22-01157-f011:**
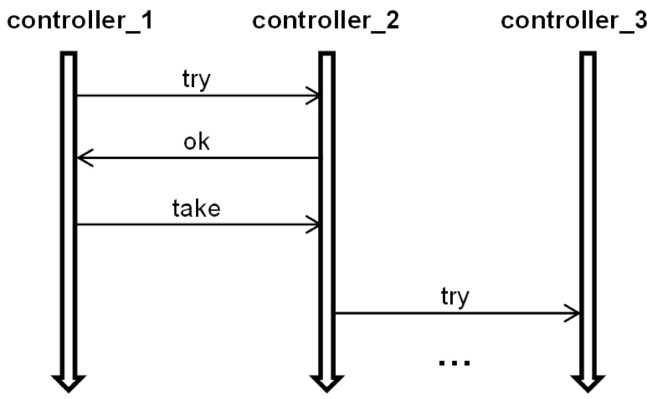
The protocol of road segment controller cooperation.

**Figure 12 sensors-22-01157-f012:**
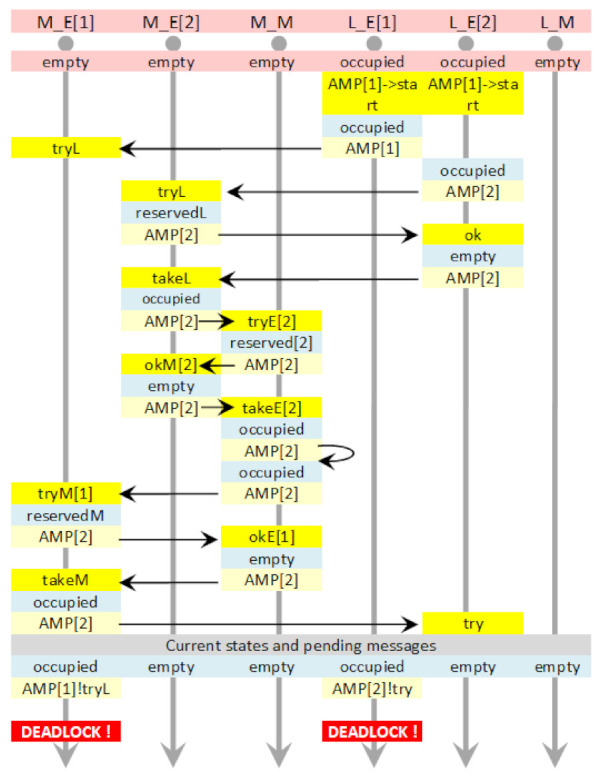
Communication structure in a trace of the timeless AMP’s behavior, leading to a deadlock.

**Figure 13 sensors-22-01157-f013:**
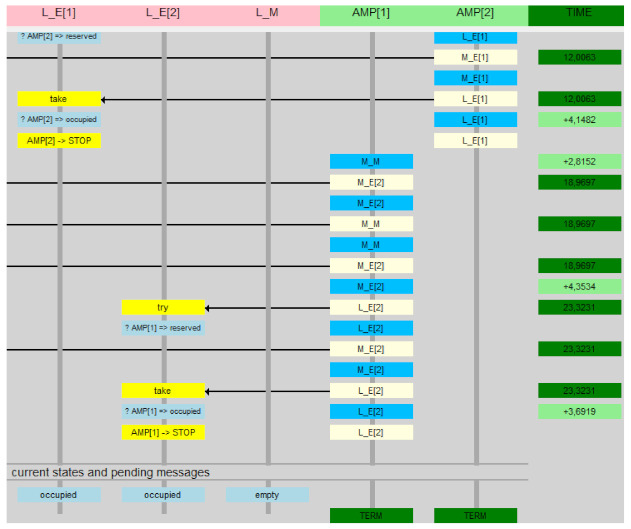
The final fragment of the witness of the corrected *AVGS* timed system.

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
