# Peer review of "Modeling and Verification of Asynchronous Systems Using Timed Integrated Model of Distributed Systems"

_sensors, 2022, doi:10.3390/s22031157_

Round 1

Reviewer 1 Report

The author introduces T-IMDS (Timed Integrated Model of Distributed Systems) which is some version of time automata and then shows how to use this model to show some properties of distributed systems. The author also claims to provide valid transformation of his model into Uppaal Time Automata. Some applications are also discussed. The results are a little bit shallow, but it could be published provided improved presentation and addition of some formal proofs.

The standard notation is practically all papers about automata uses the following pattern for automata definition: ‘A  XYZ automaton is a tuple  Axyz = (…..), where …’. Moreover simple abstract examples usually illustrate all the important definitions. Unfortunately the author not always follows this pattern which makes proper understanding unnecessarily problematic.  A definition of standard time automaton, the root of all models of this kind is not given.

There is no formal proof that a given T-IMDS is really equivalent to an Uppaal Time Automaton derived from it. Such a proof is necessary, just semi-formal reasoning is not enough.

While Petri nets are mentioned in the paper, the differences between Petri net models and time automata based models are never discussed, so it is not clear why the author has used the latter as the roots of his approach.

References seem to be rather ad hoc and a little bit random. For example LTS (Labelled Transition Systems) are much much older than the 2011 paper [29]. They are in principle Finite Automata in disguise, so their roots are in Moore, Mealy and Kleene papers from fifties, maybe the author should mention Keller’s paper from 1976 (use in concurrency). This is a minor problem but it is also easy to fix.

Summing up, the paper must be revised.

Author Response

Comment: The author introduces T-IMDS (Timed Integrated Model of Distributed Systems) which is some version of time automata and then shows how to use this model to show some properties of distributed systems. The author also claims to provide valid transformation of his model into Uppaal Time Automata. Some applications are also discussed. The results are a little bit shallow, but it could be published provided improved presentation and addition of some formal proofs.

Response - the text explaining this issue at the end of section 3.1: It should be noted that IMDS is not an automata-based model, although it can be represented as a collection of automata, in various ways, the paper [40] formally represents the conversion of IMDS to node automata or to agent automata. In this article, we use some graphical notation informally with node states as vertices and actions as transitions, because such notation is natural and makes it easier to express certain features of the IMDS. However, it should be remembered that such automata are only one of the possible interpretations; there are also others, such as the agent automata mentioned above, as well as pairs of synchronous automata (custodians and messengers [41]), or the non-automata interpretation as Petri nets shown in [21]. There is also an imperative language Rybu, compatible with IMDS, used by the student to verify their synchronization solutions [42]. Now we work on even higher level language for web service composition. Please note that IMDS is just a set of actions over the quadruple Cartesian product (MP)(MP). For programming purposes, these actions can be grouped on nodes, agents, or otherwise, which does not change the essence of the definition of formalism itself.

Comment: The standard notation is practically all papers about automata uses the following pattern for automata definition: ‘A  XYZ automaton is a tuple  Axyz = (…..), where …’. Moreover simple abstract examples usually illustrate all the important definitions. Unfortunately the author not always follows this pattern which makes proper understanding unnecessarily problematic.  A definition of standard time automaton, the root of all models of this kind is not given.

Response – IMDS is not an automata-based model, as mentioned above. The definition of Uppaal Timed Automata in Uppaal version is given in Section 4.1 (now 5.1 - syntax) and 4.2 (now 5.2 - semantics). Standard TA are useless because they do not apply urgent channels (a timed transition can be executed while a progress transition is enabled) and they do not use variables, thus no value can be passed between automata that explode the automata layout. Following the reviewer’s comment, the definitions of TA and network of TA are given in Section 2

More figures are given to illustrate the concept of T-IMDS: Fig. 1 gives the idea of timeless IMDS action. New Figure 3 presents the complete view of an action with the two actions delivering input items to it. Also, the elements of graphical illustration are numbered and referred to in the formal definition of the T-IMDS semantics. Additionally, the semantics of UTA and T-IMDS, and the translation rules are numbered instead of bullets, and they are referred to in describing the equivalence between the two formalisms and the proof of equivalence.

New Figures 6 and 7 present the two actions of the “two semaphores” example and their UTA implementation.

The example of “two-semaphores” T-IMDS system is moved to Section 5 which defines the formalism.

Comment: There is no formal proof that a given T-IMDS is really equivalent to an Uppaal Time Automaton derived from it. Such a proof is necessary, just semi-formal reasoning is not enough.

Response – the formal proof is given in the Appendix.

Comment: While Petri nets are mentioned in the paper, the differences between Petri net models and time automata based models are never discussed, so it is not clear why the author has used the latter as the roots of his approach.

Response – the new text in Section 1 While automata-based models, as well as IMDS,  can be transformed to Petri Nets [21], they are not so attractive in our opinion because it is not easy to extract the processes from the specification. We use Petri nets for the structural analysis of a verified system and some kinds of model checking [22].

Comment: References seem to be rather ad hoc and a little bit random. For example LTS (Labelled Transition Systems) are much much older than the 2011 paper [29]. They are in principle Finite Automata in disguise, so their roots are in Moore, Mealy and Kleene papers from fifties, maybe the author should mention Keller’s paper from 1976 (use in concurrency). This is a minor problem but it is also easy to fix.

Response – References cited are used in our work, and we selected them with our best knowledge. In our faculty, we try to cite the younger papers rather than the older ones. I added the citation suggested by the reviewer.

Comment: Summing up, the paper must be revised.

Response – much work has been done to follow the reviewer’s comments.

Reviewer 2 Report

  1. The abstract of this paper needs to improve in view of existing problems and their solution adopted by the new methodology.
  2. The contributions should be stressed more explicitly to let the reader immediately grasp them. From Section I, it could be seen that the authors are failed to show the limitations of the existing methods. So, the main contribution of the paper shall be highlighted and emphasized. Moreover, please cite more recent papers to make a fair literature review. In summary, redraft the introduction including background, challenges, a literature survey of recent works, research scopes, motivation, objectives, contribution, and organization of the paper.
  3. This article is not well organized, which makes it difficult to read.
  4. The process presented in this paper is so complex and not easy to understand as a general reader. So, a generic graphic is required to visualize an overview of the proposed approach, which will be explained in more detail in order that the reader can easily understand the presented concept of the implementation of the proposed approach.
  5. Finally, there are lots of grammatical errors and typos. Please thoroughly check the whole paper to correct all of these errors.

Author Response

1. Comment: The abstract of this paper needs to improve in view of existing problems and their solution adopted by the new methodology.

Response: The abstract was rewritten from scratch.

2. Comment: The contributions should be stressed more explicitly to let the reader immediately grasp them. From Section I, it could be seen that the authors are failed to show the limitations of the existing methods. So, the main contribution of the paper shall be highlighted and emphasized. Moreover, please cite more recent papers to make a fair literature review. In summary, redraft the introduction including background, challenges, a literature survey of recent works, research scopes, motivation, objectives, contribution, and organization of the paper.

Response: Section 1 was extended to cover the reviewer’s requirements.

3. Comment: This article is not well organized, which makes it difficult to read.

Response – the example of IMDS has been moved to Section 5.1 where T-IMDS is defined. More figures illustrating the T-IMDS and its UTA implementation are included. All the elements of T-IMDS and UTA in Figs. 4 and 5 (updated numbering) and new Figure 3 are numbered and referred to in translation rules. The elements of UTA semantics in Sect. 4.4 (now, 5.4), T-IMDS semantics in Sect. 5.2 (now, 4.2) and translation rules in Sect. 6.2 are numbered. Section 6.4 shows the equality of T-IMDS and its UTA implementation and in the proof of this equality in the Appendix, the references to these numbers are given.

The reviewer has not commented what was wrong with the article layout, I guess that the definitions of timeless IMDS and T-IMDS were unnecessarily separated by UTA. This was for the purpose of providing first timeless, then timed systems. Now IMDS and T-IMDS are given one after another and then the definition of UTA comes.

4. Comment: The process presented in this paper is so complex and not easy to understand as a general reader. So, a generic graphic is required to visualize an overview of the proposed approach, which will be explained in more detail in order that the reader can easily understand the presented concept of the implementation of the proposed approach.

Response – the previous response covers this issue.

5. Comment: Finally, there are lots of grammatical errors and typos. Please thoroughly check the whole paper to correct all of these errors.

Response – the text has been reviewed by a native speaker, and the author analyzed the grammar and spelling carefully.

Round 2

Reviewer 1 Report

The author has addressed all the issues I have raised. Accept as it is.

Reviewer 2 Report

No more comments.